# A comparative analysis of heterogeneity in lung cancer screening effectiveness in two randomised controlled trials

Clinical trials demonstrate that screening can reduce lung cancer mortality by over 20%. However, lung cancer screening effectiveness (reduction in lung cancer specific mortality) may vary by personal risk-factors. Here we evaluate heterogeneity in lung cancer screening effectiveness through traditional sub-group analyses, predictive modelling approaches and machine-learning in individual-level data from the Dutch-Belgian lung cancer screening trial (NELSON; 14,808 participants, 12,429 men, 2377 women, 2 persons with an unknown sex) and the National Lung Screening Trial (NLST; 53,405 participants, 31,501 men, 21,904 women). We find that screening effectiveness varies by pack-years (screening effectiveness ranges across trials: lowest groups = 26.8-50.9%, highest groups = 5.5-9.5%), smoking status (screening effectiveness ranges across trials: former smokers = 37.8-39.1%, current smokers = 16.1-22.7%) and sex (screening effectiveness ranges across trials: women = 24.6-25.3%; men = 8.3-24.9%). Furthermore, screening effectiveness varies by histology (screening effectiveness ranges across trials: adenocarcinoma = 17.8-23.0%, other lung cancers = 24.5-35.5%, small-cell carcinoma = 9.7%-11.3%). Screening is ineffective for squamous-cell carcinoma in NLST (screening effectiveness = 27.9% (95% confidence interval: 69.8% increase to 4.5% decrease) mortality increase) but effective in NELSON (screening effectiveness = 52.2% (95% confidence interval: 25.7-69.1% decrease) mortality reduction). We find that variations in screening effectiveness across pack-years, smoking status, and sex are primarily explained by a greater prevalence of histologies with favourable screening effectiveness in these groups. Our study shows that heterogeneity in lung screening effectiveness is primarily driven by histology and that relaxing smoking-related screening eligibility criteria may enhance screening effectiveness.

The National Lung Screening Trial (NLST) and Dutch-Belgian Lung Cancer Screening Trial (Nederlands–Leuvens Longkanker Screenings Onderzoek, NELSON) demonstrated that Computed Tomography (CT) screening reduces lung cancer mortality (LCM)[1,2]. Lung cancer (LC) screening effectiveness (the reduction in LCM through screening) differed between trials, with NELSON yielding a 24% LCM reduction (men) for CT screening compared to no screening, after 10 years of follow-up, while NLST found a 20% reduction for CT screening compared to chest radiography screening after 6.5 years of follow-up (16% after endpoint verification process extension at 7 years of follow-up)[3].

✉e-mail: welz@ese.eur.nl; k.tenhaaf@erasmusmc.nl

The trials differed in number of screening rounds, screening intervals and nodule management protocols. However, they also differed with regards to their participants' risk-factor prevalence and overall LC risk.

Screening effectiveness may be affected by LC risk and risk-factors like age, sex, and smoking history[1,3–8]. Furthermore, LC is a heterogeneous disease, with both screen-detectability and treatment varying across histology[3,9]. Consequently, it is uncertain whether differences in screening effectiveness are primarily driven by differences in trial designs or whether they are also affected by heterogeneity across risk-factors, LC risk or histology.

LC screening is in its implementation phase throughout Europe, spearheaded by the United Kingdom's Targeted Lung Health Checks. Determining the factors driving heterogeneity in screening effectiveness would improve identifying those most likely to benefit, aiding (cost-)effective implementation of LC screening programs in Europe. The United States Preventive Services Task Force (USPSTF) recently recommended expanding screening eligibility to younger individuals and those with fewer pack-years[10]. Moreover, the American Cancer Society (ACS) recommended against restrictions on years since smoking cessation, as it may exclude considerable numbers of high-risk individuals[11]. However, the impact of expanding LC screening eligibility criteria on screening effectiveness is unknown.

Heterogeneity in LC screening effectiveness has been traditionally evaluated through sub-groups based on singular characteristics, i.e. "one-variable-at-a-time" analyses[1,3,4]. Yet, such analyses are subject to low statistical power and multiplicity, making them prone to both false-negative and false-positive results[12]. Instead, recommendations like the Predictive Approaches to Treatment effect Heterogeneity (PATH) Statement suggest using predictive-modelling approaches[13]. Furthermore, novel machine-learning methods have been suggested to provide more accurate predictions than predictive-modelling approaches[14]. Nevertheless, some suggest novel machine-learning methods may not improve prediction and may be more prone to bias than predictive-modelling approaches[15]. In general, there is a dearth of studies comparing methodologies for assessing heterogeneity in effectiveness[16]. Hence, comparing multiple models/methodologies with common inputs (comparative modelling) is recommended to obtain more reliable conclusions[17].

In this study we evaluate the causes and magnitude of heterogeneity in LC screening effectiveness using individual-level data from NELSON and NLST through comparative modelling. Traditional sub-group analyses, predictive-modelling approaches, and machine-learning are applied to evaluate the consistency of the findings across methodologies. Models are developed in both trials; models developed in one trial are externally validated in the other. We find that screening effectiveness varies across pack-years, smoking status, and sex. However, these variations are primarily explained by a greater prevalence of histologies with favourable screening effectiveness in these groups. Overall, we show that heterogeneity in lung screening effectiveness is primarily driven by histology and that relaxing smoking-related screening eligibility criteria may enhance screening effectiveness.

## Results

### Associations between risk-factors and LCM

Associations between risk-factors and LCM were consistent between trials (Supplementary Data 1). The first stage risk- and effect-models suggest age, personal history of cancer, current smoking, cigarettes per day and years smoked were associated with increased LCM risk (Supplementary Tables S3, S4). Female sex, body-mass index, education level, and years since smoking cessation were associated with reduced LCM risk. In NLST, COPD, emphysema, LC family history, Black race and American Indian or Alaskan Native race were associated with increased LCM risk, while Asian race and Hispanic ethnicity reduced risk. Overall, risk-factors had similar effect

sizes in both trials and were consistent with risk-prediction models for LCM[18].

### Overall screening effectiveness

Figure S1 shows relative screening effectiveness estimates for overall LCM across methods. Estimates were similar across methodologies, for both NELSON (median LCM reduction across methods: 27.2%, 95% confidence interval (CI): 10.8–40.9%) and NLST (median: 15.3%, 95% CI: 3.7–25.5%) after accounting for differences in participant characteristics between trials. This may reflect differences in trial designs, such as differences in control arms (NELSON: no screening. NLST: chest radiography screening) and number of screening rounds (NELSON: four screening rounds. NLST: three screening rounds). Relative screening effectiveness was similar across baseline risk-quintiles for all risk-prediction models (Supplementary Figs. S2–S5; NELSON medians = 22.7–24.6%, NLST medians = 8.9–13.4%) and second stage risk-models (Supplementary Table S5; NELSON median = 27.4% (95% CI: 11.7–40.3%), NLST median = 15.6% (95% CI: 3.2–26.4%)). However, absolute screening benefits increased with quintile of baseline risk for all risk-prediction models (C-Outcomes Supplementary Appendix Supplementary Figs. S32–S35. NELSON LCM prevented per 1000: Quintile 1 −0.92 to 2.31 to Quintile 5: 25.8–28.3, NLST LCM prevented per 1000: Quintile 1 −0.04 to 1.03 to Quintile 5: 4.83–8.12).

### Screening effectiveness by risk-factors

Figure 1 shows the relative screening effectiveness estimates aggregated across methods by age-groups, smoking status, years since smoking cessation, accumulated pack-years, and sex (estimates by methods shown in Supplementary Figs. S6–S9; corresponding effect-models: Supplementary Tables S6–S9). Screening effectiveness varied little by age-group in both trials. In NELSON, the median estimated screening effectiveness was lower for current smokers (median = 22.7%, 95% CI: −1.8–40.0%) compared to former smokers (median = 37.8–39.0%). In NLST, differences in median estimated screening effectiveness were primarily found between current smokers and long-term quitters (≥10 years) (medians = 16.1% (95% CI: 2.4–29.9% and 39.1% (95% CI: 9.6–60.2%)). Median estimated screening effectiveness diminished with increasing pack-years (NELSON medians: <30 pack-years = 50.9%, 95% CI: 16.0–84.8%; >50 pack-years = 5.5%, 95% CI: −36.3 to 38.5%. NLST medians: 30–39 pack-years = 26.8%, 95% CI: −7.7 to 61.3%; >50 pack-years = 9.5%, 95% CI: −5.2% to 22.8%). Finally, while the median screening effectiveness in NELSON did not differ by sex (medians: females = 25.3% (95% CI: −26.9–60.1%); males = 24.9% (95% CI: 6.6–41.7%), the median screening effectiveness was greater for females in NLST (medians: females = 24.6% (95% CI: 6.7–40.6%); males = 8.3% (95% CI: −6.7–22.8%). Overall LC incidence risks estimated by risk-prediction models were lower for groups with higher estimated relative screening effectiveness in both trials (Supplementary Figs. S10, 11).

### Screening effectiveness across histologies

Figure 2 shows relative screening effectiveness varied across histologies (corresponding first-stage risk-models: Supplementary Tables S10, S11; second stage risk-models: NELSON shown in Supplementary data 2, NLST in Supplementary data 3; effect-models: Supplementary Tables S12, S13). Screening effectiveness for adenocarcinoma (median LCM reductions = 17.8–23.0%), Other lung cancers (OTH) (24.3–35.5%), and small-cell carcinoma (9.7–11.3%) were similar between trials (with and without adjustments for multiple comparisons using Benjamini-Hochberg corrections). Conversely, screening effectiveness was better for squamous-cell carcinoma (for all evaluated methods except rate-ratios) in NELSON than NLST (medians = 52.2% LCM reduction (95% CI: 25.7–69.1% decrease) versus a 27.9% increase (95% CI: 69.8% increase to 4.5% decrease)).

Like screening estimates for overall LCM, histology-specific screening effectiveness estimates were consistent across methods.

# Screening effectiveness estimates for overall LC mortality, by risk-factor

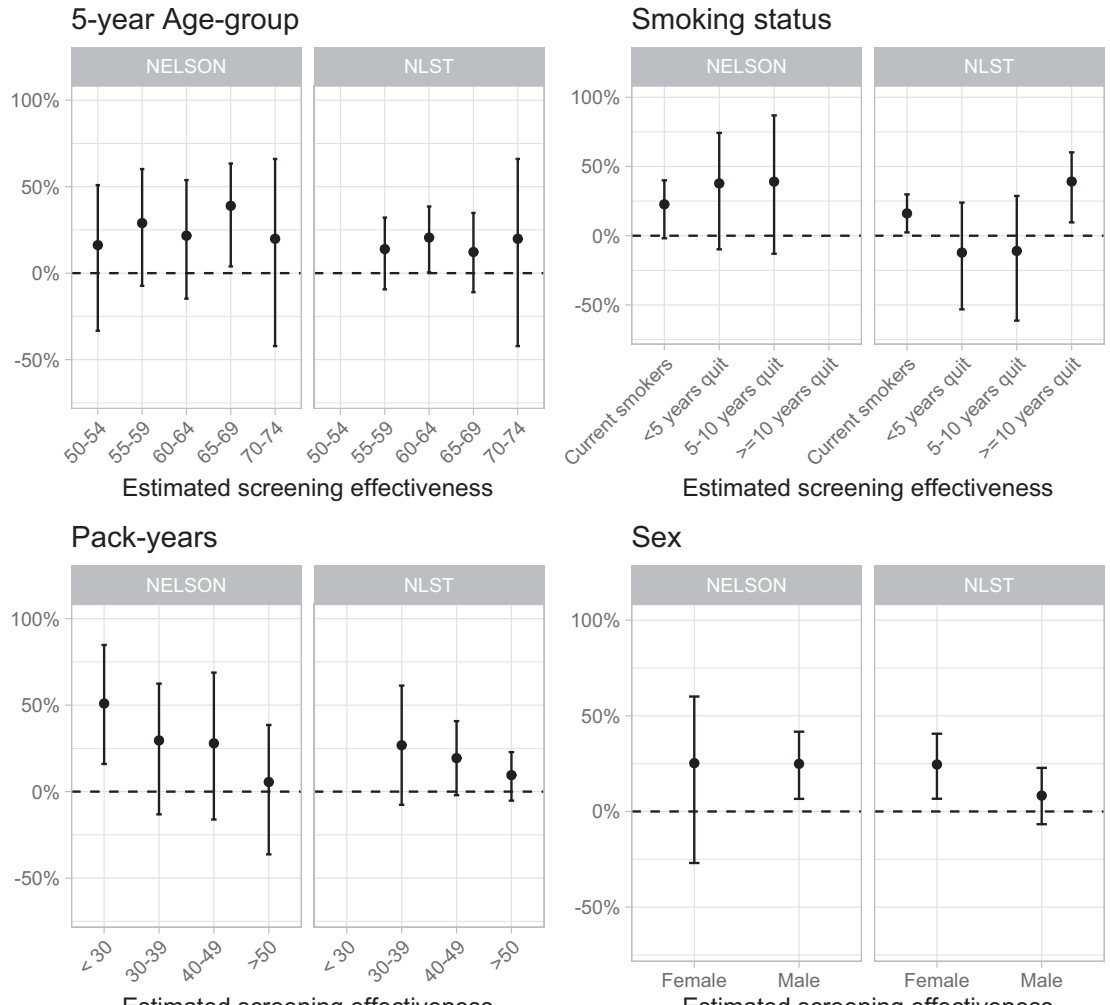

**Fig. 1 | Relative screening effectiveness for overall LCM by trial and different risk-factors aggregated across the considered methods.** Based on N = 400 lung cancer deaths in NELSON and N = 977 lung cancer deaths in NLST. The screening effectiveness estimates and 95% confidence intervals were based on the medians of the point estimates of the rate-ratios, effect models and causal forests. Similarly, the lower and upper bounds of the 95% confidence intervals were based on the medians of the lower and upper bounds across the considered methods. The estimates for the individual methods can be found in Supplementary Figs. S6–S9. Source data are provided as a Source Data file. LC Lung Cancer, LCM Lung Cancer Mortality, NLST National Lung Screening Trial, Dutch-Belgian Lung Cancer Screening Trial (NELSON Nederlands–Leuvens Longkanker Screenings Onderzoek).

However, in contrast to the heterogeneity in relative screening effectiveness found for overall LCM by smoking status, accumulated pack-years and sex, little variation was found in histology-specific effectiveness (Supplementary Figs. S12–S14; corresponding effect-models in Supplementary Tables S14–S19). This can be explained by considering the histology distributions within these groups in the CT-arms of both trials (Fig. 3). In both trials, prolonged smoking cessation, fewer pack-years, and female sex were associated with a greater prevalence of adenocarcinoma and a lower prevalence of small-cell carcinoma. Thus, variations in screening effectiveness for overall LCM found across smoking status, accumulated pack-years, and sex can be primarily explained by a greater prevalence of histologies with more favourable screening effectiveness in these groups.

### Links between overall and histology-specific effectiveness
Figure 4 demonstrates the relations between histology-specific screening effectiveness and overall screening effectiveness. Although

adenocarcinoma mortality does not show the greatest relative mortality reduction in either trial, it accounts for the greatest reduction in absolute LCM in NLST and the second greatest reduction in NELSON. Squamous-cell carcinoma accounted for both the greatest reductions in absolute and relative mortality in NELSON, while both absolute and relative mortality was increased in NLST. In NLST, the greatest relative reduction in histology-specific mortality was found for OTH, accounting for the second greatest reduction in absolute histology-specific mortality. Similarly, while OTH showed the second greatest relative reduction in histology-specific mortality in NELSON, the reduction in absolute mortality was 57% lower compared to adeno-carcinoma. Finally, for small-cell carcinoma both the relative and absolute reductions in histology-specific mortality were modest in both trials. In general, the reductions in histology-specific mortality correspond to the observed reductions in the incidence of histology-specific late-stage disease, as shown in Fig. 5, and to the stage distribution of screen-detected cases in the CT-arms (Supplementary Fig. S15).

# Screening effectiveness estimates by histology–specific mortality

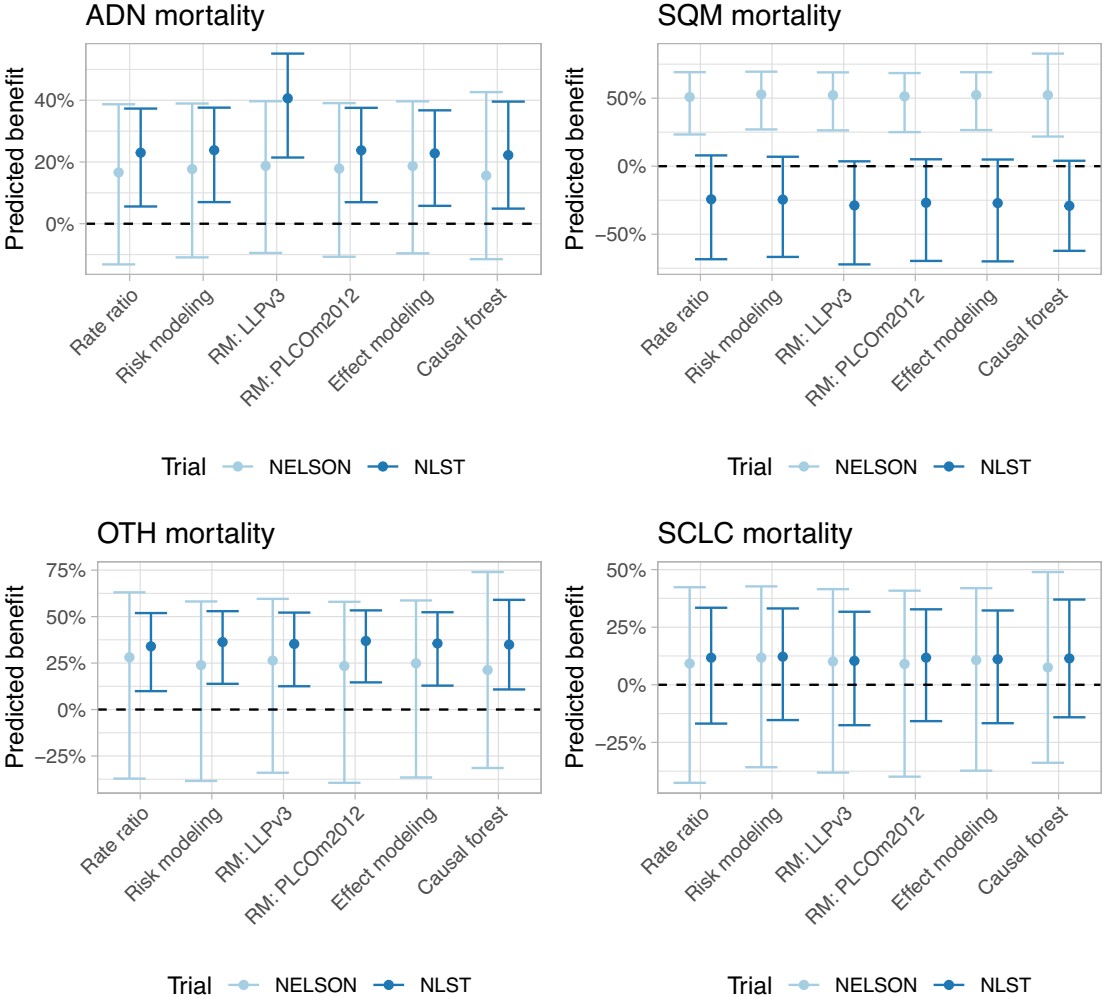

**Fig. 2 | Screening effectiveness for histology-specific mortality by methodology and trial.** Based on $N = 178$ Adenocarcinoma deaths, $N = 94$ Squamous-cell carcinoma deaths, $N = 43$ Other lung cancer deaths and $N = 84$ Small-cell carcinoma deaths in NELSON and $N = 393$ Adenocarcinoma deaths, $N = 184$ Squamous-cell carcinoma deaths, $N = 176$ Other lung cancer deaths and $N = 209$ Small-cell carcinoma deaths in NLST. The screening effectiveness estimates were based on the point estimates of each of the specific methods, by trial. The error bars reflect the 95% confidence intervals of the estimates. The adenocarcinoma-specific NLST estimate for the risk-model approach which uses LLPv3 risk in its first stage includes an interaction-effect between first-stage risk and screening effectiveness. The figure represents the estimate for the screening effectiveness parameter without the interaction effect. Source data are provided as a Source Data file. RM Risk-modelling, ADN Adenocarcinoma, SQM Squamous-cell carcinoma, OTH Other lung cancers, SCLC Small-cell carcinoma, NLST National Lung Screening Trial, Dutch-Belgian Lung Cancer Screening Trial (NELSON Nederlands–Leuvens Longkanker Screenings Onderzoek).

## C-statistics, C-for-benefit and calibration-for-benefit

Details on these outcomes are provided in the C-Outcomes Supplementary Appendix (displayed in Supplementary Figs. S17–S70 and Supplementary Tables S26–S30). In brief, overall discrimination for LCM and screening benefit were similar across methods within both development and validation sets, for both overall LCM and histology-specific LCM. Calibration for relative and absolute benefits were generally good in the development datasets for both methods, but poor in the validation sets.

## Sensitivity analyses

We evaluated whether cancers classified under "Neoplasm, malignant" and "non-small cell cancer, not otherwise specified" may have led to bias. Therefore, we evaluated the impact of assigning these cancers to one of the histology classifications ("non-small cell cancer, not otherwise specified") to non-small cell categories only based on imputations (30 total) taking into account participant characteristics, study arm and method of diagnosis (screening/clinical). In these analyses, histology-specific screening effectiveness reduced slightly for adenocarcinoma in both trials (NELSON median = 16.3% reduction (95% CI: −11.2–37.3%); NLST median = 21.6% reduction (95% CI: 5.3–35.5%)). Squamous cell screening effectiveness remained similar for NELSON (median: 52.0% reduction (95% CI: 26.8–68.7%)) but improved in NLST (median = 25.6% increase (95% CI: 66.6% increase to 5.5% decrease)). Screening effectiveness for OTH improved in both trials (NELSON median: 44.6% reduction (95% CI: −70.3–84.2%); NLST median = 52.9% reduction (95% CI: 27.6–69.9%)). Small cell screening effectiveness remained similar (NELSON median = 9.0% reduction (95% CI: −37.4–41.2%); NLST median = 11.1% reduction (95% CI: −16.7–32.6%)). Finally, we performed a pooled analysis of the individual

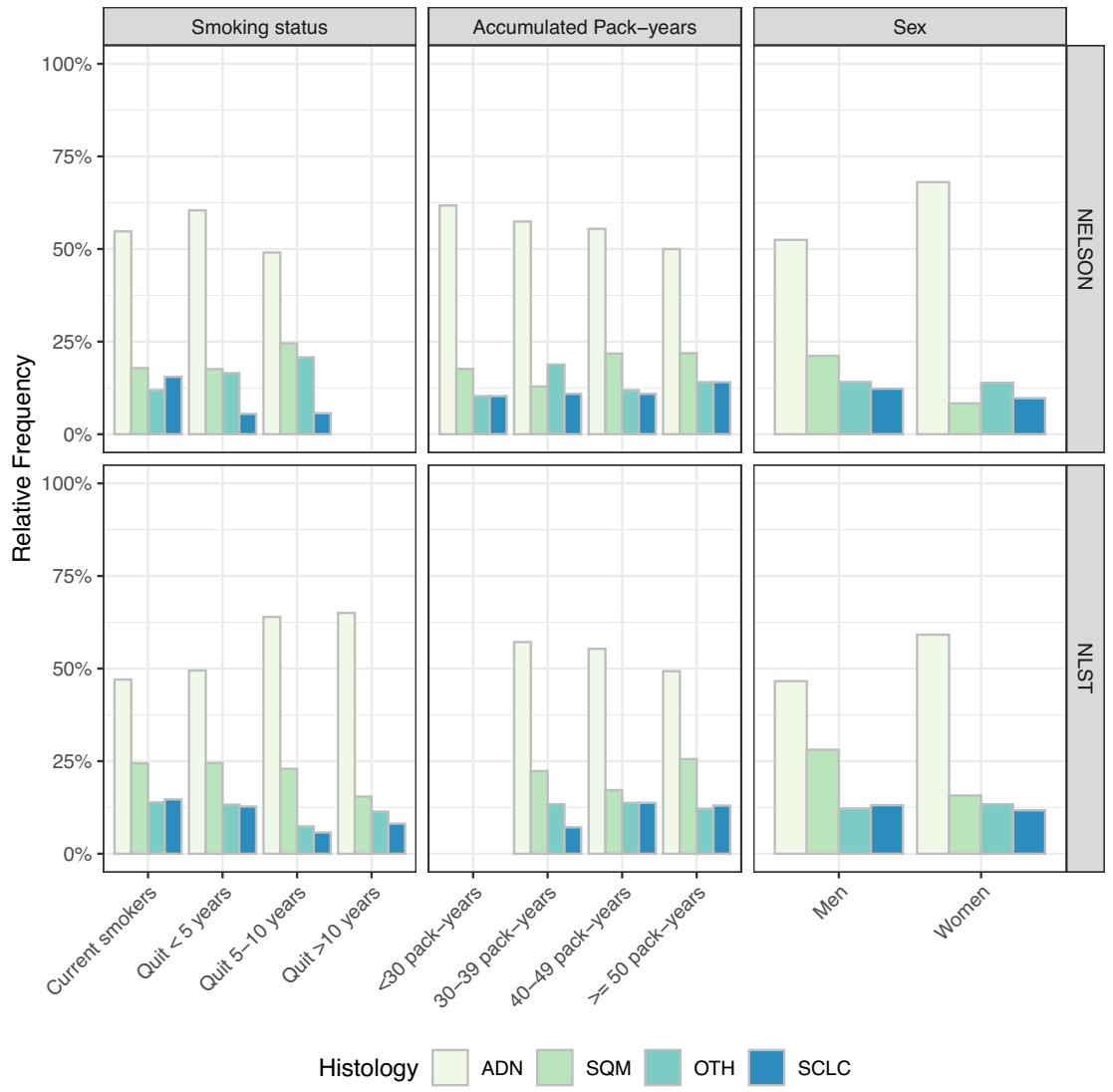

**Fig. 3 | Histology distribution by risk-factor in the CT-arms of NELSON and NLST.** Source data are provided as a Source Data file. ADN Adenocarcinoma, SQM Squamous-cell carcinoma, OTH Other lung cancers, SCLC Small-cell carcinoma, NLST National Lung Screening Trial, Dutch-Belgian Lung Cancer Screening Trial (NELSON Nederlands–Leuvens Longkanker Screenings Onderzoek).

patient-level data from NELSON and NLST to further evaluate pre-screening differences between trials, such as differences in eligibility criteria and trial population. This model produced equivalent results as the models in the main analyses, suggesting they appropriately account for pre-screening differences between trials (Supplementary Material: Pooled analysis, Supplementary Figs. S71–S72 and Supplementary data 4).

## Discussion

Heterogeneity in LC screening effectiveness was evaluated in individual-level data from the two largest screening trials through traditional sub-group analyses, predictive-modelling and machine-learning. Estimates were similar across methodologies, for both NELSON (median LCM reduction across methods = 27.2%, 95% CI: 10.8–40.9%) and NLST (median = 15.3%, 95% CI: 3.7–25.5%) after accounting for differences in participant characteristics between trials. The greater reduction in LCM in NELSON compared to NLST may reflect differences in trial designs, such as differences in control arms (NELSON: no screening. NLST: chest radiography screening) and

number of screening rounds (NELSON: four screening rounds. NLST: three screening rounds). Screening effectiveness diminished with increasing pack-years (LCM reductions across trials = 26.8–50.9% in the lowest pack-year groups compared to 5.5–9.5% in the highest pack-year groups), former smokers compared to current smokers (LCM reductions = 37.8–39.1% versus 16.1–22.7%) and women compared to men (LCM reductions = 24.6–25.3% versus 8.3–24.9%). LC risks estimated by risk-prediction models were lower for groups with higher estimated relative screening effectiveness. However, histology was identified as the main effect modifier of heterogeneity in LC screening effectiveness. Screening was effective for adenocarcinoma and OTH, reducing mortality by 17.8–23.0% (medians NELSON&NLST) and 24.5–35.5% (medians NELSON&NLST), respectively. In contrast, screening was less effective for small-cell cancers, reducing mortality by 9.7–11.3% (medians NELSON&NLST) and discordant results were found for squamous-cell carcinoma (NELSON: median reduction of 52.2% (95% CI: 25.7–69.1% decrease); NLST: median increase of 27.9% (95% CI: 69.8% increase to 4.5% decrease)). Our findings are consistent with natural-history models that estimate longer preclinical durations

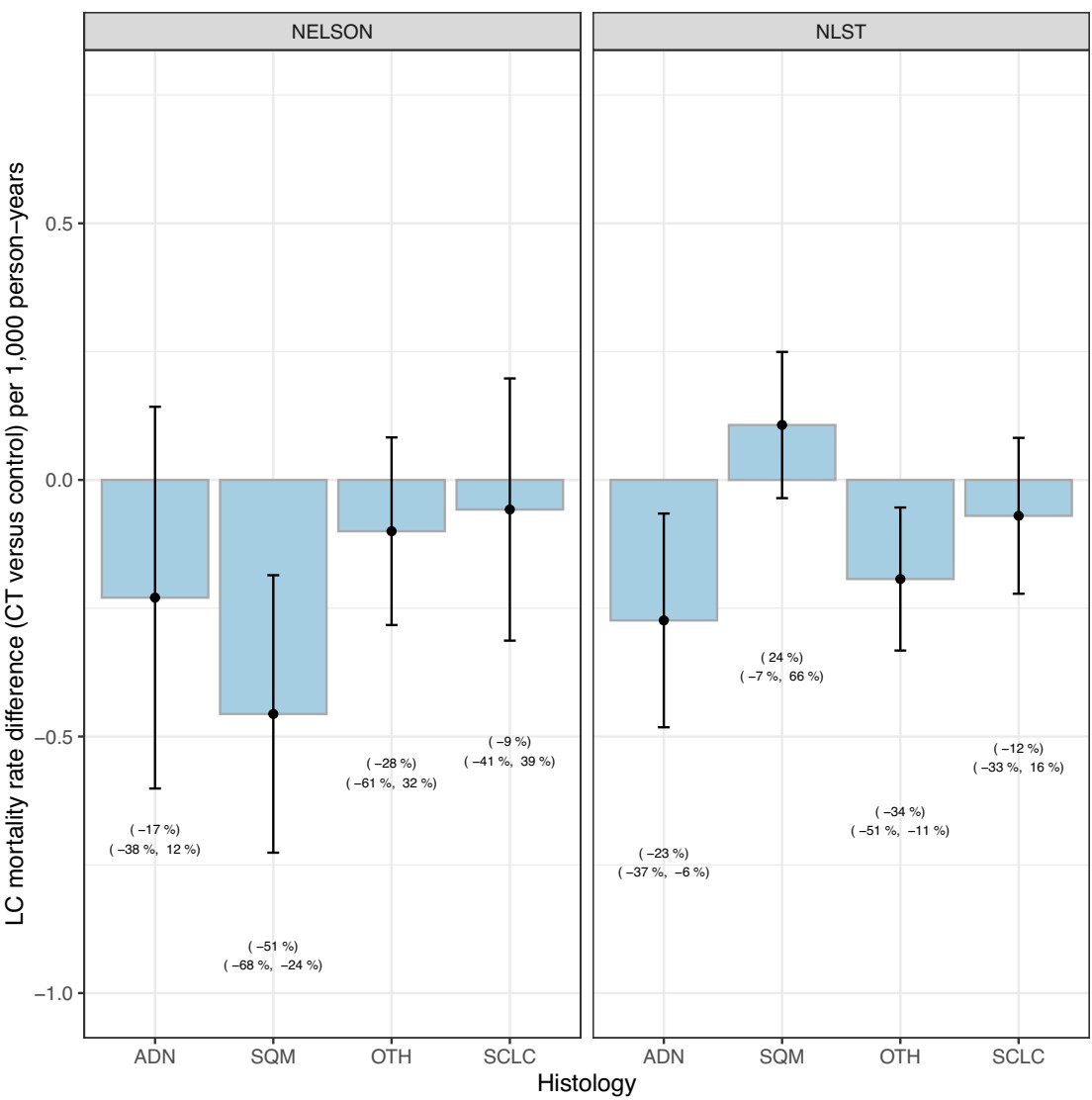

**Fig. 4 | Histology-specific lung cancer mortality rate differences in NELSON and NLST.** Based on $N = 178$ Adenocarcinoma deaths, $N = 94$ Squamous-cell carcinoma deaths, $N = 43$ Other lung cancer deaths and $N = 84$ Small-cell carcinoma deaths in NELSON and $N = 393$ Adenocarcinoma deaths, $N = 184$ Squamous-cell carcinoma deaths, $N = 176$ Other lung cancer deaths and $N = 209$ Small-cell carcinoma deaths in NLST. The lung cancer mortality rate differences represent the difference in lung cancer mortality rate per 1000 person-years in the CT arm compared to the control arm by histology in each trial. The error bars reflect the 95% confidence intervals of the mortality rate differences. The percentages below the mortality rate differences represent the relative differences in mortality rates compared to the control-arm, along with their 95% confidence intervals. Source data are provided as a Source Data file. ADN Adenocarcinoma, SQM Squamous-cell carcinoma, OTH Other lung cancers, SCLC Small-cell carcinoma, NLST National Lung Screening Trial, Dutch-Belgian Lung Cancer Screening Trial (NELSON Nederlands–Leuvens Longkanker Screenings Onderzoek).

and greater screen-detectability for histologies for which we find greater screening effectiveness[9,19]. In particular, we find greater screening effectiveness for histologies that predominantly develop in locations that allow for easier detection through screening. For example, adenocarcinomas develop predominantly in the periphery of the lungs, as opposed to small-cell cancers that tend to be centrally located[20,21]. Furthermore, our findings are consistent with regards to observed relations between histology and smoking behaviour, and variations in survival by histology and smoking behaviour[22–27]. Consequently, these mechanisms may drive the heterogeneity in screening effectiveness found in our study.

Although squamous-cell carcinoma incidence has decreased, it still represents over 20% of LC[28]. While analyses based on NLST have suggested screening may not be beneficial for squamous-cell carcinoma, our analyses suggest it was beneficial in NELSON[3]. This may be

in part due to differences in nodule management protocols. Semi-automated measurements of nodule volume and volume doubling time as applied in NELSON has been shown to be more accurate in detecting nodule growth than the manual measurements of nodule diameter used in NLST[29]. This is supported by a recent review that demonstrates that there may not be a significant change in volume at a three month follow-up scan, even when the volume doubling time is less than 400 days[30]. Consequently, future studies should evaluate the impact of the differences in nodule management protocols between the trials on histology-specific screening effectiveness.

Screening effectiveness was greater for women, those with fewer accumulated pack-years and former smokers, due to the higher prevalence of histologies for which screening effectiveness was greater. Thus, the 2021 USPSTF recommendation to lower the minimum pack-years for screening eligibility and the ACS recommendation to relax

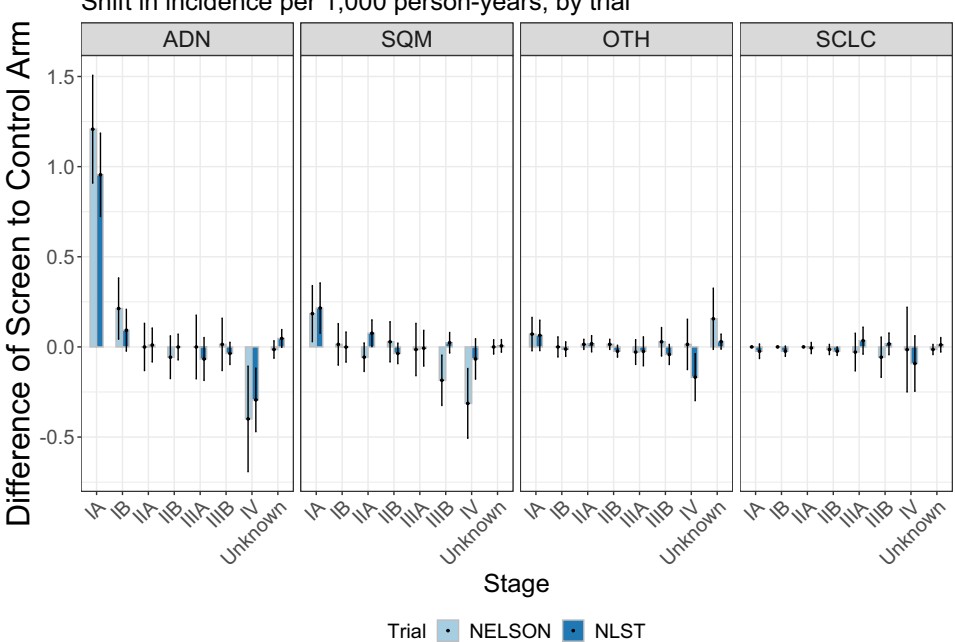

**Fig. 5 | Differences in lung cancer incidence rates between the screen and control arms of NELSON and NLST by histology and stage.** Based on $N = 372$ Adenocarcinomas, $N = 174$ Squamous-cell carcinomas, $N = 93$ Other lung cancers and $N = 103$ Small-cell carcinoma in NELSON and $N = 985$ Adenocarcinomas, $N = 461$ Squamous-cell carcinomas, $N = 298$ Other lung cancers and $N = 287$ Small-cell carcinomas in NLST. The lung cancer incidence rate differences represent the difference in lung cancer incidence rate per 1000 person-years in the CT arm compared to the control arm by stage and histology in each trial. The error bars reflect the 95% confidence intervals of the incidence rate differences. Source data are provided as a Source Data file. ADN Adenocarcinoma, SQM Squamous-cell carcinoma, OTH Other lung cancers, SCLC Small-cell carcinoma, NLST National Lung Screening Trial, Dutch-Belgian Lung Cancer Screening Trial (NELSON Nederlands–Leuvens Longkanker Screenings Onderzoek).

restrictions on the numbers of years since smoking cessation will improve eligibility among individuals in whom histologies with greater screening effectiveness are more prevalent[10,11,31]. These relaxations have also been shown to improve eligibility among individuals of African-American ancestry who are more likely to develop squamous-cell carcinoma compared to individuals of European ancestry[28,32,33]. Consequently, lung cancer screening effectiveness for African-Americans should be further evaluated. Our analyses suggest a potential relation between family history of LC and adenocarcinoma (Supplementary Tables S11 and S15). This is of particular importance for regions with high LC incidence in never-smokers, whom predominantly develop adenocarcinomas. Consequently, studies investigating screening in never-smokers should further evaluate the impact of heterogeneity in screening effectiveness by histology[34].

Integrating smoking cessation support has been shown to enhance the effectiveness of LC screening in reducing LCM through reducing the risk for developing LC. Our analyses suggest that integrating smoking cessation support further enhances the effectiveness of LC screening in reducing LCM through two additional pathways. Firstly, our analyses suggest that screening effectiveness is greater for former smokers compared to current smokers, particularly for long-term former smokers. Secondly, successful smoking cessation prevents the further accumulation of additional pack-years, which our analyses suggest is associated with reduced screening effectiveness. Consequently, the findings of our study may be used to further improve the uptake of smoking cessation services in LC screening programs.

Our study suggests that relaxing eligibility criteria improves selection of individuals in whom histologies with greater screening effectiveness are more prevalent. However, this also expands eligibility to lower-risk individuals. Consequently, criteria relaxations may yield diminishing returns in additional deaths prevented and reduce screening efficiency (screens required per LC detected). Currently ongoing implementation efforts, like the United Kingdom's targeted lung health checks, are predominantly focused on regions with high LC rates to optimize screening efficiency and available healthcare resources[35,36]. However, as these areas are more likely to be populated with high-risk (heavier smoking) individuals, LC screening effectiveness could be lower than anticipated. While studies indicate that extending screening to lower risk individuals may be cost-effective in the U.S., this requires additional health-care resources[37]. Consequently, full evaluations of the trade-offs between screening effectiveness, efficiency, health inequities and required health-care resources are essential to guide implementation efforts. Furthermore, studies should evaluate how information on heterogeneity in screening effectiveness can be included and impact shared decision-making processes.

Previous studies evaluated heterogeneity in LC screening effectiveness[3,5–8]. Wille indicated greater effectiveness for individuals with Chronic Obstructive Pulmonary Disease who smoked ≥35 pack-years[5]. Infante suggested greater effectiveness for those with <40 pack-years, current smokers and a Forced Expiratory Volume in 1 s (FEV1)% ≥ 80[8]. Nevertheless, these studies evaluated trials with non-significant results and should be interpreted with caution[13]. Pinsky found heterogeneity in screening effectiveness by histology and potential heterogeneity by sex in NLST[3]. However, their analysis considered one-variable-at-a-time rather than predictive approaches, precluding the identification of effect modification by patient characteristics. Our study confirms heterogeneity in screening effectiveness between histologies, but also accounted for confounding and effect modification by patient characteristics. For example, Pinsky et al. found greater screening effectiveness for current smokers compared to former smokers. Our results are consistent when smoking status alone is considered (Supplementary Fig. S16).

However, our results indicate greater benefits for long-term former smokers when time since smoking cessation is also taken into consideration, demonstrating the importance of including sufficient granularity in former smoking behaviour. Furthermore, our study expands on these findings by identifying groups in which histologies with greater screening effectiveness are more prevalent, providing important guidance to clinicians. Kovalchik evaluated screening effectiveness across risk-groups in NLST[6]. Similarly to our findings, they found relative screening effectiveness was constant across risk-groups and absolute effectiveness increased with risk, however their approach did not evaluate screening effectiveness across different risk-factors nor consider histology. Kumar evaluated the cost-effectiveness of risk-based screening in the NLST through a multi-state model[7]. Similarly to Kovalchik, they found absolute screening effectiveness increased with risk, but did not consider histology[6]. Our results indicate that while screening effectiveness does not vary by overall risk, it does vary across individual components of risk. Therefore, future studies should not only consider overall LC(M) risk, but also consider individual components of risk with sufficient granularity.

In contrast to previous studies, we evaluated heterogeneity in LC screening effectiveness in two trials with statistically significant results. We performed a comprehensive analysis that accounted for participant risk-factors, LC risk, and histology. The definitions of the risk-factors were aligned between trials and showed similar effect sizes in both trials. We considered both relative and absolute effectiveness through various methods. We equalized post-screening follow-up and accounted for differences in participant characteristics between trials.

We evaluated different methods with different strengths and limitations, as outlined in Supplementary Table S2. These strengths vary from straightforward interpretation (one-variable-at-a-time, risk-prediction models), explicitly accounting for different covariates (predictive modelling and machine-learning approaches) to not requiring assumptions for linearity (machine learning). However, they are also subject to limitations such as no or limited accounting for other covariates (one-variable-at-a-time, risk-prediction models), not allowing interactions between screening effectiveness and covariates (risk-modelling), overfitting (effect-modelling) or non-straightforward interpretation (machine-learning). However, despite the differences in their underlying assumptions, differences in strengths and limitations, the results were consistent across methodologies, demonstrating the robustness of our findings. Furthermore, we evaluated calibration and discrimination, which are often poorly reported for both predictive-modelling and machine-learning approaches[13,15].

Our findings are based on two trials. While overall screening effectiveness was greater in NELSON than NLST after accounting for participant characteristics and post-screening follow-up, the trials also differed in number of screening rounds and screening interval lengths[1,2]. Furthermore, while NELSON compared CT screening to no screening, NLST compared CT screening to chest-radiography screening[1,2]. Nodules detected at incidence screening rounds vary in LC risk from those detected at baseline, which may affect screening effectiveness across screening rounds[38]. Furthermore, it is uncertain whether there were differences in LC treatment patterns between the trials. Finally, while NELSON compared CT screening to no screening, NLST compared CT screening to chest-radiography screening. The Prostate, Lung, Colorectal, and Ovarian (PLCO) Cancer Screening Trial found a non-significant LCM reduction of 9% for chest-radiography screening[39]. Consequently, CT screening effectiveness in NLST may be underestimated. However, smoking exposure in PLCO was lower (20% of PLCO participants were NLST-eligible; 45% were never-smokers), which our investigation suggests would lead to greater prevalence of LC with favourable screening effectiveness. This is supported by comparing the NLST and PLCO chest-radiography arms LC histology distributions (Supplementary Table S20). Thus, chest-radiography screening in PLCO may have been more effective than NLST due to a greater prevalence of LC with favourable screening effectiveness. Hence, future studies should evaluate whether and how differences in trial designs, nodule growth patterns and LC treatments affected screening effectiveness across histologies.

Participants of both trials were more likely to be younger and have ceased smoking, but were generally representative of the general population meeting their inclusion criteria[40,41]. Still, it is well known that the individuals eligible for lung cancer screening are more likely to have comorbidities such as COPD than those included in the trials[42,43]. These comorbidities increase the overall risk of lung cancer, reduce life-expectancy and may affect both treatment effectiveness and the histological type of lung cancer that develops[18,22,23,44–49]. Thus, future research should further evaluate the interplay between comorbid conditions and screening effectiveness.

Our analysis was limited to 4–4.5 years post-screening to equalize follow-up between trials. Consequently, our estimates should be interpreted as representative for the evaluated follow-up period in both trials. However, while the number of life-years gained through screening may increase with prolonged post-screening follow-up, the effect on LCM may be diluted. Indeed, this was shown in extended follow-up analyses of NLST, although dilution was modest[50]. However, limited information on histology was available for cancers detected during the extended follow-up process (<8%), precluding evaluation of heterogeneity in screening effectiveness by histology. The analyses considered population characteristics at baseline. However, 10–24% of current smokers at baseline in the trials ceased smoking post-enrollment[51–53]. Consequently, this may have affected the estimates of smoking cessation on LC screening effectiveness.

Misclassification of histology can occur and recommendations for the pathological classification of LC have changed over time[54]. However, sensitivity analysis regarding LC misclassification similar to Pinsky, did not affect our findings with regards to variations in histology-screening effectiveness and prevalence of histologies across risk-factors[3]. Furthermore, we applied penalised estimation to mitigate the potential for overfitting as well as non-parametric methods. Given the consistency of our findings across different methodologies, the consistency between the risk-factors included in our models and those included in well validated risk-prediction models for LC(M), and the good calibration performance of the models, we believe the potential for model misspecification to be modest. Targeted therapy and immunotherapy use was limited during the trials, but their uptake has increased considerably since then. In addition, histology-specific incidence has changed in past decades and may change further[28]. Thus, the effects of the increased uptake of novel therapies and changes in histology-specific incidence on LC screening effectiveness should be monitored.

Overall, our study shows that heterogeneity in LC screening effectiveness is primarily driven by histology. The 2021 USPSTF and 2023 ACS guidelines are more likely to include individuals with higher prevalence of histologies with high screening effectiveness compared to their previous guidelines, due to relaxation of smoking-related eligibility criteria. Integrating risk-reduction interventions in LC screening programs may further enhance screening effectiveness.

## Methods

### Trial oversight

The NELSON trial (Netherlands Trial Register number NL580) was approved by the Dutch Minister of Health and the medical ethics committee at each participating site. The NLST (ClinicalTrials.gov number NCT00047385) was approved by the institutional review board at each of the 33 participating medical institutions. Participants in both trials provided written informed consent before randomisation.

## Study population

An overview of the trials is provided in Supplementary Table S1[1,2]. In brief, the NELSON trial enrolled, from December 2003 through July 2006, individuals between the ages of 50–74, who smoked at least 15 cigarettes per day for ≥25 years or 10 cigarettes per day for ≥30 years and were current smokers or former smokers who quit <10 years ago. The intervention arm received four rounds of computed tomography screening with different intervals: at baseline, year 1 (1-year interval), year 3 (2-year interval) and year 5.5 (2.5-year interval). The participants in the control arm received no screening. The NLST enrolled, from August 2002 through April 2004, individuals between the ages of 55–74, who smoked at least 30 pack-years and were current smokers or former smokers who quit <15 years ago. The intervention arm received three rounds of computed tomography screening with a one-year interval each: at baseline, year 1 and year 2. The participants in the control arm received three rounds of chest radiography screening with the same schedule as the intervention arm. For Belgian NELSON participants, only summarised information on LC incidence and mortality was available. Consequently, they were excluded (NELSON: $n = 934$). Furthermore, inadvertently randomised never-smokers (NELSON: $n = 17$), individuals diagnosed with LC before randomisation (NELSON: $n = 10$), and individuals without known LC incidence dates (NELSON: $n = 23$, NLST: $n = 47$) were excluded. The final analyses included 14,808 NELSON (12,429 men, 2377 women, 2 persons with an unknown sex) and 53,405 NLST participants (31,501 men, 21,904 women). The median age of the participants was 58 in NELSON and 60 in NLST. In NELSON, LC diagnoses ($n = 743$) and LC deaths ($n = 400$) occurring between randomisation and December 31st 2015, or 10 years of follow-up since randomisation (whichever came first) were considered. In NLST, LC diagnoses ($n = 2055$) and LC deaths ($n = 977$) occurring between randomisation and 7 years of follow-up were considered. For NLST, information on stage and histology was derived through the follow-up of individuals with positive screens, annual status update and medical records, including pathology and tumour staging reports, for all suspected LC[2]. For NELSON, data on cancer diagnosis, histology and stage, vital status, and cause of death were obtained through linkages with the Dutch Center for Genealogic and Heraldic Studies, Statistics Netherlands, and the Dutch Cancer Registry[1]. In NELSON, randomization by screening center was applied. In NLST, blocked randomization stratified by age, sex, and screening center was applied, with blocks of length six or eight (with order of assignment being random within each block). Overall screening adherence rates were similar across the trials (NELSON: 90%, NLST: 95%)[1,2]. Risk-factor data were collected through epidemiologic questionnaires administered at study entry and harmonised across trials (Data Harmonisation Supplementary Appendix, Supplementary Tables S21, S22). The risk-factors considered in developing and validating the different (modelling) methods are shown in Supplementary Table S2. The analyses considered both sexes (with sex as defined by biological attributes; based on self-reported data). We refer the reader to the original trial protocols of the NELSON and NLST trials for the reporting of sex-disaggregated data. Sex was considered as an explanatory variable in all applied modelling methods.

## Statistics & reproducibility

**Screening effectiveness.** The outcome assessed by the models was screening effectiveness, which was defined as the reduction in LCM achieved through CT screening. LCM was evaluated rather than survival, as survival estimates for screen-detected cases may be affected by lead-time and length-time bias[55,56]. Screening effectiveness was evaluated by comparing the difference in LCM between each trial's screening and control group (NELSON: no screening; NLST: chest radiography screening) from randomisation until the end of each trial's respective follow-up period since randomization (10 years in NELSON and 7 years in NLST). Heterogeneity in screening effectiveness

was evaluated through traditional sub-group analyses, predictive-modelling approaches and machine-learning.

Traditional sub-group analyses considered one-variable-at-a-time analyses, in which the trial population is serially divided into different groups (e.g. men and women) and the difference in LCM between groups is evaluated through exact univariable rate ratio tests based on the Poisson distribution. In addition, effectiveness across risk-levels by different risk-prediction models, namely PLCOm2012 and LLPv3 (Risk-Prediction Model Supplementary Appendix, Supplementary Tables S23–S25)[45].

Predictive-modelling approaches, as suggested by the PATH statement, considered risk-modelling and effect-modelling, evaluated using penalised (elastic net) logistic regressions applying 10-fold cross-validation (Statistical Methodology Supplementary Appendix)[13]. The penalised regressions applied 10-fold cross-validation to determine the tuning parameters using the algorithms of Friedman et al.[57]. In brief, risk-modelling applies a two-stage approach. The first-stage model estimates the baseline LCM risk (as a linear predictor) of an individual without including information on whether they received screening. The second-stage model estimates the LCM risk by including both information on whether the person received screening, as well as an interaction term between the linear predictor derived from the first-stage model and screening assignment. Effect models attempt to model screening effect heterogeneity explicitly by including both baseline characteristics as well as interaction terms between baseline characteristics and screening assignment.

Risk-modelling approaches also evaluated substituting the baseline-risk estimates with estimates from the aforementioned risk-prediction models to provide insights into potential variation in screening effectiveness across LC incidence risk. In the risk-modelling approaches, screening effectiveness by risk interactions were tested through likelihood-ratio tests. Effect-modelling approaches evaluated a limited number of interaction-effects due to their high potential for overfitting (even when true interaction-effects are present)[58]. Interactions between sex and screening effectiveness was considered, as both LC preclinical duration (and thus the potential for screen-detection) and screening effectiveness have been suggested to differ by sex and histology[1,3,4,9]. In addition, interactions between age and screening effectiveness, and pack-years smoked and screening effectiveness were considered to evaluate the impacts of the 2021 USPSTF recommendations versus their 2013 recommendations[10,31]. Finally, screening effectiveness by smoking status and years since smoking cessation were considered to evaluate the impacts of the recent ACS recommendations[11]. The machine-learning approach considered causal forests, which define the overall screening effectiveness as the average predicted screening effect across all individuals[14]. Each forest was estimated using 2000 trees.

Supplementary Table S2 provides an overview of the considered methods, while full descriptions of each methodology and how estimands were harmonised to provide relative estimates for LCM reduction are provided in the Statistical Methodology Supplementary Appendix. All methods, except the rate-ratios, account for differences in participant characteristics between trials[13]. Thus, the screening effectiveness parameter in these models represents the effect of screening on reducing LCM after accounting for confounders. For each applicable method, calibration-for-benefit (through calibration plots), concordance-statistic for LCM (C-statistic) and concordance-statistic for benefit (C-for-benefit) were evaluated in accordance with the PATH statement[13,59]. In brief, calibration-for-benefit demonstrates how well models predict the absolute or relative benefit for an individual, while the C-statistic&C-for-benefit demonstrate whether models can distinguish those who die from LC/benefit from screening from those who do not, respectively. The C-for-benefit does not have an established benchmark, but generally ranges between 0.40 and 0.60[59]. For all methods, heterogeneity by quintile of the method's estimated baseline

risk distribution was assessed. Each method was used to derive relative and absolute screening effectiveness estimates by baseline risk quintile (chosen to allow comparison to previous studies) within each trial[6]. For methods capable of external validation, models developed in NELSON were externally validated (i.e. without model recalibration) in NLST and vice versa. R version 4.2.0 and R-package predmod (version 1.0.0) were used for the analyses[60]. For the one-variable-at-a-time approaches, continuous variables were categorized to create groups. For the predictive modelling and machine-learning approaches, continuous variables were kept continuous while a linear functional relationship was assumed. However, for comparisons with the one-variable-at-a-time approaches, models with similarly categorized versions of these variables were used.

It is not known which histology a person will develop. However, LC survival and screen-detectability varies across histology[9]. Thus, evaluating whether there is heterogeneity in screening effectiveness by histology is important for both clinical and public health decision-making. Therefore, we evaluated screening effectiveness by histology-specific mortality, similar to Pinsky[3]. An overview of the histology-specific mortality is provided in Supplementary data 1.

**Missing data.** The overall level of missing data was 4.5% in NELSON and 0.5% in NLST. Overall, missing data across participant characteristics varied from 0 to 3% (Supplementary data 1). However, race/ethnic group (NELSON: not inquired: NLST: <2%), BMI (NELSON: 4–5%; NLST: -1%), COPD and emphysema (NELSON: 35% missing due to not being asked during the first recruitment round:NLST: <1%), and family history of LC (NELSON: not inquired; NLST: <2%) had higher rates of missing data. These variables are primarily used in the risk-prediction models; as such we believe the impact of these missing data to be modest. Missing data were handled through multiple imputation ($k$-nearest-neighbour approach) using R-package "VIM" (version 6.2.2). All candidate predictor variables (including the LCM outcome variable) were included in the imputation and the number of nearest neighbours ($k$) was set to 5. Analyses were performed using 30 imputations, pooled through Rubin's rules.

### Reporting summary
Further information on research design is available in the Nature Portfolio Reporting Summary linked to this article.

## Data availability
The data generated to create the main and Supplementary Figs. are available in the Source Data file. Access to the deidentified participant data from the National Lung Screening Trial can be obtained for research purposes from the United States National Cancer Institute Cancer Data Access Center at https://biometry.nci. nih.gov/cdas. Access to the deidentified participant data from the Dutch-Belgian lung cancer screening trial can be obtained for research purposes through the NELSON data access board (https://umcgresearch.org/w/nelson-dataset). Source data are provided with this paper.

## Code availability
The Predmod package used to perform the analyses in this study can be found at: https://github.com/mwelz/predmod.

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

## Acknowledgements

The NELSON trial was supported by the Netherlands Organisation of Health Research and Development, the Dutch Cancer Society (KWF Kankerbestrijding), the Health Insurance Innovation Foundation (Innovatiefonds Zorgverzekeraars), G.Ph. Verhagen Stichting, the Rotterdam Oncologic Thoracic Study Group, the Erasmus Trust Fund, Stichting tegen Kanker, Vlaamse Liga tegen Kanker and Lokaal Gezondheids Overleg (LOGO) Leuven. Siemens Germany provided four workstations and software for volume measurements. This work was supported by TRANSCAN-2 Joint Transnational Call 2016 (JTC 2016) project code

TRANSCAN-045 CLEARLY (H.J.d.K.), a Convergence (Erasmus MC, Erasmus University Rotterdam and Delft University of Technology) Open Mind grant (Convergence for Individualising TReatment Using Statistical approaches (CITRUS)) (M.W., A.A., K.t.H.), a VENI grant from the Dutch Research Council/ Netherlands Organisation of Health Research (ZonMW) (grant number 09150161910060) (K.t.H.) and a VIDI grant from the Dutch Research Council (grant number VI.Vidi.195.141) (A.A.). We thank the NELSON and NLST participants for their contributions to this study. We also thank dr. M.A. den Bakker (Maasstad Hospital Rotterdam) for their advice regarding the lung cancer histology classifications. We thank the National Cancer Institute (NCI) for access to NCI's data collected by the NLST. The statements contained herein are solely those of the authors and do not represent or imply concurrence or endorsement by NCI.

## Author contributions

M.W.: conceptualisation, data curation, investigation, formal analysis, methodology, software, writing – original draft. C.M.vdA.: data curation, investigation, writing – review & editing. A.A.: supervision, investigation, writing – review & editing. A.A.N.: supervision, investigation, writing – review & editing. M.H.: investigation, writing – review & editing. H.J.M.G.: investigation, writing – review & editing. P.A.d.J.: investigation, writing – review & editing. J.A.: investigation, writing – review & editing. M.O.: investigation, writing – review & editing. H.J.d.K.: conceptualisation, funding acquisition, supervision, writing – review & editing. K.t.H.: conceptualisation, data curation, investigation, formal analysis, funding acquisition, methodology, software, supervision, writing – original draft.

## Competing interests

C.M.v.d.A. declares roles in the WHO-IARC European Code Against Cancer working group, B3care user committee and Expert Mission Cancer Screening in Georgia outside of the submitted work. H.J.G. declares consulting fees from Eli Lilly outside of the submitted work. P.A.d.J. research support from Philips Healthcare to their institute outside of the submitted work. J.A. declares speakers fees from Eli Lilly, MSD and BIOCAD, patents from Pamgene and Amphara, data safety monitoring/advisory board participation for Eli-Lilly, Amphara, BIOCAD and MSD, boardmembership of the IASLC and stock ownership in Amphera outside of the submitted work. H.J.d.K. declares consulting fees from Bayer and honoraria from TEVA/Menarini/Astra Zeneca outside of the submitted work. K.t.H. declares grants from NIH, Horizon 2020, University of Zurich, Cancer Research UK, Cancer Australia and the Australian Ministry of Health, speakers fees from Johnson&Johnson and Centre Hospitalier Universitaire Vaudois paid to their institute and travel support from the Rescue Lung Society outside of the submitted work. The remaining authors declare no competing interests.

## Additional information

**Max Welz**[1,2] ✉, **Carlijn M. van der Aalst**[1], **Andreas Alfons** ®[2], **Andrea A. Naghi**[2,3], **Marjolein A. Heuvelmans**[4,5,6], **Harry J. M. Groen** ®[7], **Pim A. de Jong**[8], **Joachim Aerts** ®[9], **Matthijs Oudkerk** ®[5], **Harry J. de Koning** ®[1,10], **Kevin ten Haaf** ®[1,10] ✉, **On behalf of the NELSON trial consortium**

[1]Department of Public Health, Erasmus MC–University Medical Center Rotterdam, Rotterdam, the Netherlands. [2]Econometric Institute, Erasmus University Rotterdam, Rotterdam, the Netherlands. [3]Department of Business Analytics and Applied Economics, School of Business and Management, Queen Mary University of London, London, UK. [4]University of Groningen. University Medical Center Groningen, Department of Epidemiology, Groningen, the Netherlands. [5]Institute for Diagnostic Accuracy, Groningen, the Netherlands. [6]Department of Respiratory Medicine, Amsterdam University Medical Center, Amsterdam, the Netherlands. [7]Rijksuniversiteit Groningen, Groningen, the Netherlands. [8]Department of Radiology, University Medical Center Utrecht, Utrecht, the Netherlands. [9]Department of Pulmonary Medicine, Erasmus MC–University Medical Center Rotterdam, Rotterdam, the Netherlands. [10]These authors contributed equally: Harry J. de Koning, Kevin ten Haaf. ✉e-mail: welz@ese.eur.nl; k.tenhaaf@erasmusmc.nl

## the NELSON trial consortium

**Carlijn M. van der Aalst**[1], **Marjolein A. Heuvelmans**[2,3,4], **Harry J. M. Groen**[5], **Pim A. de Jong**[6], **Joachim Aerts**[7], **Matthijs Oudkerk**[3], **Harry J. de Koning**[1] & **Kevin ten Haaf**[1]

