## [Transparent Peer Review file · Nature Communications]

A comparative analysis of heterogeneity in lung cancer screening effectiveness in two randomised controlled trials

Corresponding Author: Dr Kevin ten Haaf

Version 0:

Reviewer comments:

Reviewer #1

(Remarks to the Author)

This paper uses individual participant data from the two largest controlled trials of lung cancer screening to examine heterogeneity of treatment effect (HTE), particularly the effect of lung cancer histology. Using subgroup analysis, risk modeling and machine learning-based approaches, the authors found that female sex, fewer pack years and former smoking were associated with greater effectiveness. Interestingly, adenocarcinoma histology was associated with greater effectiveness in both trials, while squamous cell histology was associated with more effective screening in the (European) NELSON trial and less effective screening in the (American) National Lung Screening Trial (NLST).

Strengths include the clinical relevance of the research question, the resourceful re-use of data from clinical trials to perform secondary analysis, the multiple statistical and artificial intelligence-based approaches, and a few provocative findings. The paper is ambitious and technically impressive.

The main limitation is that the paper tries to do too many things at once. The main headline, that histology may be the primary driver of HTE, gets a bit lost when also comparing and contrasting results from the two trials and across the different methods. Similarly, there is only cursory discussion of why results might differ between trials and little or no mention of the strengths and limitations of the different methodological approaches. Most important, the discrepancy in the results regarding the directionality of the effect of squamous cell histology is not explored adequately.

Importantly, the authors make strong claims regarding causation that may not be supported by the data. While adenocarcinoma histology is indeed associated with fewer pack years and female sex, additional analyses would be necessary to show that AC histology is mediating the relationships between pack years and screening effectiveness or sex and screening effectiveness. In addition, the counterintuitive finding that squamous histology was strongly associated with screening effectiveness in NELSON does not seem to support this hypothesized mediator effect, because squamous histology is typically strongly associated with greater pack years and male sex. There is a bit of hand waving in the Discussion about the effects of "histology" (not otherwise specified) that elides this difficulty.

The finding of greater effectiveness of screening among participants with fewer pack-years and former smoking histories is also counter-intuitive and not entirely consistent with prior studies. Kovalchik et al. found that the relative effectiveness of screening in the NLST was largely constant across risk groups, and therefore, that effectiveness on the absolute scale increases as lung cancer risk increases. Although the p values for the interaction terms were not statistically significant in an earlier secondary analysis of NLST outcomes by Pinsky and colleagues, their group found that screening was more effective (in relative terms) among participants that currently smoked. Again, there is a bit of hand waving in the Discussion on this point instead of a more thoughtful wrestling with a discrepancy that does not seem minor.

Lastly, the analysis does not take into account the role of comorbid conditions and how they interact with lung cancer risk, histologic type, screening effectiveness and competing risks of death. There is an extensive emerging literature on this topic that is not cited or discussed.

Minor comments:

Figure 1: this can be moved to the supplement. Instead, it would be helpful to include a Table with key characteristics of participants in the two trials.

Figure 2: how were results pooled across methods?

Figure 3: the outlier effect of adenocarcinoma histology when using the LLP model is potentially interesting and unsettling. What are some possible explanations?

Reviewer #2

(Remarks to the Author)

The authors present a detailed comparison of the 2 largest positive lung cancer screening trials. Their findings are fairly presented and discussed and the conclusions are very largely supported by their findings. In addition, their discussion of the implications of their work and the trade-offs implied in the potential changes they suggest is valuable. A particular strength of the work is the comprehensive approach to the comparison, employing subgroup analyses, predictive modelling and machine learning approaches. The consistency of the findings is notable and adds confidence to the conclusions.

The main determinant of screening effectiveness is the histological type of the lung cancer. Adenocarcinoma is more effectively detected than small cell lung cancer. This is discussed well between lines 286 and 294. Whilst various factors are considered, such as preclinical duration and smoking behaviour, location of the tumour is not discussed. This is an omission as adenocarcinoma tends to be peripheral and hence easily detected, whilst small cell lung cancer is central and can be more challenging to detect.

The anomaly of squamous carcinoma detection effectiveness variance between the 2 studies is marked and attributed speculatively to semi-automated protocols being used in one study. This seems unlikely to account for such a large variance, which is worthy of further investigation. Is there any potential for confusion with non-malignant lesions, which might be better discriminated by automated protocols?

The authors make important suggestions for further work to investigate hypotheses generated in this work.

A few minor niggles:

Lines 58-60 could usefully be rephrased.

Lines 171-174 discuss an important finding and it would be useful to have topline data in the main paper.

Lines 260-268 contains speculation on the results, more appropriate for the discussion.

Lines 407-414 make an important point, although the logical path from their findings should be clarified.

Reviewer #3

(Remarks to the Author)

This is an analysis of over 60,000 patients, pooled from two randomized screening trials (NELSON, NSLT), with the objective to assess the impact of patient heterogeneity on lung cancer screening effectiveness. Both standard approaches statistical methods and machine learning algorithms were considered to predict lung cancer mortality risk.

Overall, the manuscript is hard to follow, as insufficient details are provided in the main text to understand the structure and goals of analyses and evaluate the findings. In general, I recommend to adhere more closely to TRIPOD statement guidelines and checklist (<https://www.tripod-statement.org/resources/>), but I provide specific comments below.

1. Please provide in the "Study population" section a detailed description of the arms of NELSON and NSLT, including interventions and evaluated screening strategies.
2. Better define in the methods the outcome that is predicted by all models/machine learning algorithms. "Lung cancer mortality" is too vague. How was this assessed? What is the time horizon for mortality predictions?
3. Please explain or provide references explaining why the evaluation of LCM is not affected by lead-time/length-time bias.
4. The "one-variable-at-the-time" analysis approach should be described in more detail in the main text. The Supplement describes provides a (unnecessarily?) detailed technical description, but it is just based on exact univariable rate ratio tests based on the Poisson distribution.
5. The definition of screening effectiveness should be clarified. In particular, it is not clear what is the specific effect that is being estimated (screening vs. no screening?) and what is the affect outcome variable (the risk predictions from the different models/algorithms? Why?)

Version 1:

Reviewer comments:

Reviewer #1

(Remarks to the Author)

The authors have addressed all of my questions. I very much appreciated their comprehensive and thoughtful response!

Reviewer #2

(Remarks to the Author)

The authors have addressed the reviewers' comments carefully and in detail. The changes they have made to the paper have significantly strengthened it.

Reviewer #3

(Remarks to the Author)

I thank the Authors for addressing my comments in a satisfactory manner. I have no further feedback to provide.

We thank the reviewers for their thoughtful comments. In the next paragraphs we describe how we addressed their comments and incorporated their suggestions.

REVIEWER COMMENTS

Reviewer #1 (Remarks to the Author):

This paper uses individual participant data from the two largest controlled trials of lung cancer screening to examine heterogeneity of treatment effect (HTE), particularly the effect of lung cancer histology. Using subgroup analysis, risk modeling and machine learning-based approaches, the authors found that female sex, fewer pack years and former smoking were associated with greater effectiveness. Interestingly, adenocarcinoma histology was associated with greater effectiveness in both trials, while squamous cell histology was associated with more effective screening in the (European) NELSON trial and less effective screening in the (American) National Lung Screening Trial (NLST).

Strengths include the clinical relevance of the research question, the resourceful re-use of data from clinical trials to perform secondary analysis, the multiple statistical and artificial intelligence-based approaches, and a few provocative findings. The paper is ambitious and technically impressive.

We thank the reviewer for their kind comments.

The main limitation is that the paper tries to do too many things at once. The main headline, that histology may be the primary driver of HTE, gets a bit lost when also comparing and contrasting results from the two trials and across the different methods. Similarly, there is only cursory discussion of why results might differ between trials and little or no mention of the strengths and limitations of the different methodological approaches. Most important, the discrepancy in the results regarding the directionality of the effect of squamous cell histology is not explored adequately.

We believe that the comprehensive approach taken in our paper is essential: as mentioned by reviewer #2 the consistency of our findings is valuable in adding confidence to the conclusion of our investigation. We will further demonstrate this in our response to reviewer #1's query on the consistency with findings from prior studies and will address the directionality of the squamous cell histology in a later comment of the reviewer.

Although the strengths and limitations of the different approaches are outlined in Supplementary Table S2, we agree with the reviewer that adding additional discussion in the main text would be valuable to the reader. We have integrated this in the discussion as follows:

“We evaluated different methods with different strengths and limitations, as outlined in Supplementary Table S2. These strengths vary from straightforward interpretation (one-variable-at-a-time, risk-prediction models), explicitly accounting for different covariates (predictive modelling and machine-learning approaches) to not requiring assumptions for linearity (machine learning). However, they are also subject to limitations such as not or limited accounting for other covariates (one-variable-at-a-time, risk-prediction models), not allowing interactions between screening effectiveness and covariates (risk-modelling), overfitting (effect-modeling) or non-straightforward interpretation (machine-learning).

However, despite the differences in their underlying assumptions, differences in strengths and limitations, the results were consistent across methodologies, demonstrating the robustness of our findings.”

Furthermore, we have added additional discussion on differences between the trials, supported by an additional supplemental table (Supplementary Table S22):

“Our findings are based on two trials. While overall screening effectiveness was greater in NELSON than NLST after accounting for participant characteristics and post-screening follow-up, the trials also differed in number of screening rounds and screening interval lengths. Nodules detected at incidence screening rounds vary in LC risk from those detected at baseline, which may affect screening effectiveness across screening rounds. Furthermore, it is uncertain whether there were differences in LC treatment patterns between the trials. Finally, while NELSON compared CT screening to no screening, NLST compared CT screening to chest-radiography screening. The Prostate, Lung, Colorectal, and Ovarian (PLCO) Cancer Screening Trial found a non-significant LCM reduction of 9% for chest-radiography screening. Consequently, CT screening effectiveness in NLST may be underestimated. However, smoking exposure in PLCO was lower (20% of PLCO participants were NLST-eligible; 45% were never-smokers), which our investigation suggests would lead to greater prevalence of LC with favourable screening effectiveness. This is supported by comparing the NLST and PLCO chest-radiography arms LC histology distributions (Supplementary Table S22). Thus, chest-radiography screening in PLCO may have been more effective than NLST due to a greater prevalence of LC with favourable screening effectiveness. Hence, future studies should evaluate whether and how differences in trial designs, nodule growth patterns and LC treatments affected screening effectiveness across histologies. ”

Importantly, the authors make strong claims regarding causation that may not be supported by the data. While adenocarcinoma histology is indeed associated with fewer pack years and female sex, additional analyses would be necessary to show that AC histology is mediating the relationships between pack years and screening effectiveness or sex and screening effectiveness.

We thank the reviewer for raising this point. Please note that we intend to state that the effects of sex and pack years on screening effectiveness are largely explained by histology, i.e., that these effects are small when controlling for histology. To clarify this, we rephrased our statement in the abstract (after describing findings on how screening effectiveness varies by pack years and sex, among other factors):

“Heterogeneity in LC screening effectiveness is primarily driven by histology.”

We have performed additional analyses that further demonstrate that histology drives the relationship between screening effectiveness and pack-years/sex as follows. For each trial, we evaluated three types of effect-models (all estimated through penalized regression accounting for multiple imputation with 30 imputations as done in the main analyses):

- 1) Effect models that consider histology as a covariate, and interactions between screening effectiveness and each of the considered histological groupings.
- 2) Effect models that consider histology as a covariate, and alongside the interactions between screening effectiveness and the histological groupings also consider an interaction between screening effectiveness and sex.

- 3) Effect models that consider histology as a covariate, and alongside the interactions between screening effectiveness and the histological groupings also consider an interaction between screening effectiveness and pack-years.

The tables below show the beta coefficients for each of these respective models by trial. Table 1 shows the effect models that consider interactions between screening effectiveness and each of the considered histological groupings. From the parameter estimates, it can be noted that including histology as a covariate provides considerable information on who was diagnosed with lung cancer and thus who is at risk of dying from lung cancer. Thus, this precludes the use of models that include histology as a covariate for decision-making for screening eligibility as this is not known before diagnosis. Consequently, this is why we chose an approach similarly to Pinsky et al in our main analyses.

Table 2 shows the effect models that include both interactions between screening effectiveness and the different histologies, as well as the interaction between screening effectiveness and sex. As can be seen from the parameter estimates, the interaction between screening effectiveness and sex is negligible compared to the interactions between screening effectiveness and histology. Similarly, Table 3 shows the same for the interaction between screening effectiveness and pack-years compared to the interactions between screening effectiveness and histology. Consequently, this demonstrates that histology is the primary driver of screening effectiveness. We have included this in the manuscript as follows:

“Thus, variations in screening effectiveness for overall LCM found across smoking status, accumulated pack-years, and sex can be primarily explained by a greater prevalence of histologies with more favourable screening effectiveness in these groups.”

and

“Analyses that considered interaction effects between screening effectiveness and the different histologies yielded consistent results with regards to the variation in histology-specific screening effectiveness by accumulated pack-years and sex.”

Table 1: Parameter estimates for the effect-model approaches for overall LCM (with interactions between screening effectiveness and histological groupings)

Risk-factors	NELSON parameter estimates	NLST parameter estimates
Intercept	-24.290	-24.282
Age (per 1-year increase)	0.012	0.024
Female sex* (binary)	-0.495	-0.356
Personal history of cancer (binary)	-0.389	0.098
Body-mass index, per 1 unit increase	0.033	0.003
Level of education, per 1 unit increase	-0.040	-0.048
Current smoking status** (binary)	0.199	0.363
Number of cigarettes per day, per 1 unit increase	0.002	0.006
Number of years smoked, per 1 unit increase	0.015	-0.013
Number of years since smoking cessation, per 1 unit increase	-0.004	-

Chronic obstructive pulmonary disease (binary)	-	-0.112
Presence of emphysema (binary)	-	0.170
Family history of lung cancer (binary)	-	-0.023
Asian race/ethnic group*** (binary)	Not inquired in NELSON	-0.218
Black race/ethnic group*** (binary)	Not inquired in NELSON	-0.056
Hispanic race/ethnic group*** (binary)	Not inquired in NELSON	-0.722
American Indian or Alaskan Native race/ethnic group*** (binary)	Not inquired in NELSON	0.784
Native Hawaiian or Pacific Islander race/ethnic group*** (binary)	Not inquired in NELSON	1.689
Adenocarcinoma histology (binary)	22.689	23.264
Squamous histology (binary)	22.575	22.626
Other histology (binary)	22.854	23.820
Small cell histology (binary)	23.399	24.163
Screening effectiveness for Adenocarcinoma histology (binary)	-1.131	-0.874
Screening effectiveness interaction with Squamous histology (binary) ****	0.281	0.997
Screening effectiveness interaction with Other histology (binary) ****	-0.334	0.298
Screening effectiveness interaction with Small cell histology (binary) ****	1.468	0.854

Table notes: *compared to male sex, **compared to former smokers, ***compared to a White race/ethnic group **** because of the perfect predictor problem (dummy trap) Adenocarcinoma is the base category. Hence for the non-adenocarcinoma histologies the full parameter estimate should be obtained through addition of the adenocarcinoma screening effectiveness parameter estimate.

Table 2: Parameter estimates for the effect-model approaches for overall LCM (with interactions between screening effectiveness and histological groupings and screening effectiveness and sex)

Risk-factors	NELSON parameter estimates	NLST parameter estimates
Intercept	-24.323	-24.317
Age (per 1-year increase)	0.013	0.024
Female sex* (binary)	-0.413	-0.294
Personal history of cancer (binary)	-0.390	0.095
Body-mass index, per 1 unit increase	0.034	0.003
Level of education, per 1 unit increase	-0.040	-0.048
Current smoking status** (binary)	0.203	0.363
Number of cigarettes per day, per 1 unit increase	0.002	0.006
Number of years smoked, per 1 unit increase	0.015	-0.013
Number of years since smoking cessation, per 1 unit increase	-0.004	-
Chronic obstructive pulmonary disease (binary)	-	-0.116
Presence of emphysema (binary)	-	0.171
Family history of lung cancer (binary)	-	-0.025
Asian race/ethnic group*** (binary)	Not inquired in NELSON	-0.220

Black race/ethnic group*** (binary)	Not inquired in NELSON	-0.052
Hispanic race/ethnic group*** (binary)	Not inquired in NELSON	-0.728
American Indian or Alaskan Native race/ethnic group*** (binary)	Not inquired in NELSON	0.779
Native Hawaiian or Pacific Islander race/ethnic group*** (binary)	Not inquired in NELSON	1.696
Adenocarcinoma histology (binary)	22.688	23.263
Squamous histology (binary)	22.579	22.633
Other histology (binary)	22.853	23.815
Small cell histology (binary)	23.392	24.157
Screening effectiveness for Adenocarcinoma histology (binary)	-1.103	-0.823
Screening effectiveness interaction with Squamous histology (binary) ****	0.267	0.979
Screening effectiveness interaction with Other histology (binary) ****	-0.336	0.302
Screening effectiveness interaction with Small cell histology (binary) ****	1.474	0.857
Screening effectiveness interaction with female sex (binary)	-0.155	-0.121

Table notes: *compared to male sex, **compared to former smokers, ***compared to a White race/ethnic group **** because of the perfect predictor problem (dummy trap) Adenocarcinoma is the base category. Hence for the non-adenocarcinoma histologies the full parameter estimate should be obtained through addition of the adenocarcinoma screening effectiveness parameter estimate.

Table 3: Parameter estimates for the effect-model approaches for overall LCM (with interactions between screening effectiveness and histological groupings and screening effectiveness and packyears)

Risk-factors	NELSON parameter estimates	NLST parameter estimates
Intercept	-24.12	-24.376
Age (per 1-year increase)	0.01	0.024
Female sex* (binary)	-0.49	-0.364
Personal history of cancer (binary)	-0.39	0.091
Body-mass index, per 1 unit increase	0.03	-
Level of education, per 1 unit increase	-0.04	-0.047
Current smoking status** (binary)	0.19	0.361
Number of cigarettes per day, per 1 unit increase	-	0.009
Number of years smoked, per 1 unit increase	0.01	-0.011
Number of years since smoking cessation, per 1 unit increase	-0.01	-
Chronic obstructive pulmonary disease (binary)	-	-0.106
Presence of emphysema (binary)	-	0.171
Family history of lung cancer (binary)	-	-0.220
Asian race/ethnic group*** (binary)	Not inquired in NELSON	-0.066
Black race/ethnic group*** (binary)	Not inquired in NELSON	-0.759

Hispanic race/ethnic group*** (binary)	Not inquired in NELSON	0.797
American Indian or Alaskan Native race/ethnic group*** (binary)	Not inquired in NELSON	1.686
Native Hawaiian or Pacific Islander race/ethnic group*** (binary)	Not inquired in NELSON	-0.106
Adenocarcinoma histology (binary)	22.69	23.264
Squamous histology (binary)	22.57	22.615
Other histology (binary)	22.85	23.820
Small cell histology (binary)	23.39	24.160
Screening effectiveness for Adenocarcinoma histology (binary)	-1.11	-0.811
Screening effectiveness interaction with Squamous histology (binary) ****	0.29	1.021
Screening effectiveness interaction with Other histology (binary) ****	-0.37	0.296
Screening effectiveness interaction with Small cell histology (binary) ****	1.50	0.860
Screening effectiveness interaction with <30 accumulated pack-years (binary)	-0.23	Participants with <30 pack-years were not included in NLST
Screening effectiveness interaction with 30-39 accumulated pack-years (binary)	0.12	-
Screening effectiveness interaction with 40-49 accumulated pack-years (binary)	-0.22	0.110
Screening effectiveness interaction with ≥50 accumulated pack-years (binary)	0.09	-0.151

Table notes: *compared to male sex, **compared to former smokers, ***compared to a White race/ethnic group **** because of the perfect predictor problem (dummy trap) Adenocarcinoma is the base category. Hence for the non-adenocarcinoma histologies the full parameter estimate should be obtained through addition of the adenocarcinoma screening effectiveness parameter estimate.

In addition, the counterintuitive finding that squamous histology was strongly associated with screening effectiveness in NELSON does not seem to support this hypothesized mediator effect, because squamous histology is typically strongly associated with greater pack years and male sex. There is a bit of hand waving in the Discussion about the effects of "histology" (not otherwise specified) that elides this difficulty.

We believe the finding on the association between the squamous histology and screening effectiveness in NELSON can be further explained by the difference in detectability between overall tumour growth and volume doubling time. Jiang et al (European Journal of Cancer, 2024) recently evaluated the lung cancer growth by histology for tumours with an initial volume between 100-300 mm³ as shown in Table 4 from their paper (presented below).

Table 4 from Jiang et al (European Journal of Cancer, 2024)

Table 4

Lung cancer growth estimation over time based on pooled mean VDT from initial volumes of 100–300 mm³.

Lung cancer type	VDT (days)	3 month follow-up		1 year follow-up		2 year follow-up	
		Folds	Volume (mm ³)	Folds	Volume (mm ³)	Folds	Volume (mm ³)
Solid	207	1.35	135 – 406	3.39	339 – 1018	11.52	1152 – 3457
Adenocarcinoma	223	1.32	132 – 397	3.11	311 – 933	9.67	967 – 2900
Squamous cell carcinoma	140	1.56	156 – 468	6.07	607 – 1822	36.90	3690 – 11069
Small cell lung cancer	73	2.34	234 – 702	31.45	3145 – 9434	988.82	98882 – 296647
Other lung cancer	178	1.42	142 – 426	4.15	415 – 1245	17.23	1723 – 5169
Part-solid	536	1.12	112 – 337	1.60	160 – 481	2.57	257 – 772
Nonsolid	669	1.10	110 – 329	1.46	146 – 438	2.13	213 – 639

This estimation started from initial tumor volumes of 100-300 mm³ (as defined for indeterminate nodules by the NELSON study). VDT refers to the pooled mean VDT derived from our meta-analysis. Folds represent the fold increase in tumor volume over the initial volume. $Folds = 2^{Time\ interval/VDT}$. Volume provides the range of tumor volumes measured at the end of each period. $Volume = Initial\ volume\ (100-300\ mm^3) \times Folds$. Bold numbers highlight significant and timely volume changes over the specified periods. VDT, volume doubling time.

Both NELSON and NLST evaluated indeterminate nodules of a specific size when detected through a follow-up scan 3-6 months later rather than immediate referral for additional diagnostics: nodules with a volume of 100-300 mm³ (NELSON) or a diameter of 4-10 mm (NLST) respectively. Based on the results of the follow-up scan, participants were then referred to further follow-up. In NELSON, nodules with a Volume Doubling Time (VDT) of <400 days were referred while in NLST nodules with a growth of ≥10% in diameter were referred. As the table from Jiang et al indicate, at the 3 month follow-up scan, the volume (and thus size) of most histologies (except small cell carcinoma) may not significantly change, but the VDT for solid cancers is less than 400 days. Therefore, the inclusion of VDT in the NELSON protocol may have contributed to a better detectability of squamous cell carcinoma in NELSON compared to NLST.

We have included this as follows in the discussion:

“While analyses based on NLST have suggested screening may not be beneficial for squamous-cell carcinoma, our analyses suggest it was beneficial in NELSON. This may be in part due to differences in nodule management protocols. Semi-automated measurements of nodule volume and volume doubling time as applied in NELSON has been shown to be more accurate in detecting nodule growth than the manual measurements of nodule diameter used in NLST. This is supported by a recent review that demonstrates that there may not be a significant change in volume at a three month follow-up scan, even when the volume doubling time is less than 400 days. Consequently, future studies should evaluate the impact of the differences in nodule management protocols between the trials on histology-specific screening effectiveness.”

In addition, analyses on the volume doubling time by histology that corrected for characteristics such as smoking behaviour demonstrate that while smoking influences the overall risk and type of histology that develops, these characteristics do not affect tumor growth within individual histological subtypes (Adler, Lung Cancer, 2002). This further supports our results indicating that histology is the main driver of screening effectiveness. We have included this as follows in the discussion:

“Furthermore, our findings are consistent with regards to observed relations between histology and smoking behaviour, smoking behaviour and tumour growth within histologies and variations in survival by histology and smoking behaviour.”

The finding of greater effectiveness of screening among participants with fewer pack-years and former smoking histories is also counter-intuitive and not entirely consistent with prior studies. Kovalchik et al. found that the relative effectiveness of screening in the NLST was largely constant across risk groups, and therefore, that effectiveness on the absolute scale increases as lung cancer risk increases. Although the p values for the interaction terms were not statistically significant in an earlier secondary analysis of NLST outcomes by Pinsky and colleagues, their group found that screening was more effective (in relative terms) among participants that currently smoked. Again, there is a bit of hand waving in the Discussion on this point instead of a more thoughtful wrestling with a discrepancy that does not seem minor.

The reviewer highlights different points for further clarification, namely: 1) the comparison of our results to the results of Kovalchik et al with regards to relative and absolute effectiveness by lung cancer risk; 2) the comparison of our results to those of Pinsky et al with regards to current and former smokers; and 3) the overall impact of screening effectiveness among those with fewer pack-years and former smoking histories. In the next paragraphs, we will address each of these points.

1) Comparison to Kovalchik et al. The reviewer indicates Kovalchik et al found that relative screening effectiveness in the NLST was constant across risk-groups. Indeed, we similarly find that relative screening effectiveness was similar across baseline risk-quintiles for all risk-prediction models (Supplementary Figures S2-S5) and second stage risk-models (Supplementary Table S5) in both NLST and NELSON. We similarly confirm their finding that effectiveness on the absolute scale increases as lung cancer risk increases, as demonstrated in C-Outcomes Supplementary Appendix Figures C16-C19.

We have further clarified these findings in the results section through the following adaptations (incorporating a later comment by reviewer #2 as well):

“Relative screening effectiveness was similar across baseline risk-quintiles for all risk-prediction models (Supplementary Figures S2-S5; NELSON medians: 22.7-24.6%, NLST medians: 8.9-13.4%) and second stage risk-models (Supplementary Table S5; NELSON median: 27.4%, NLST median: 15.6%). However, absolute screening benefits increased with quintile of baseline risk for all risk-prediction models (C-Outcomes Supplementary Appendix Figures C16-C19 NELSON LCM prevented per 1,000: Quintile 1 -0.92 to 2.31 to Quintile 5: 25.8-28.3, NLST LCM prevented per 1,000: Quintile 1 -0.04 to 1.03 to Quintile 5: 4.83-8.12).”

And have added the following paragraph to the discussion to highlight the consistency between our results and those of Kovalchik:

“Kovalchik evaluated screening effectiveness across risk-groups in NLST. Similarly to our findings, they found relative screening effectiveness was constant across risk-groups and absolute effectiveness increased with risk, however their approach did not evaluate screening effectiveness across different risk-factors nor consider histology.”

2) Comparison to Pinsky et al. The reviewer indicates that in a secondary analysis of NLST outcomes by Pinsky and colleagues, screening in relative terms was found to be more effective among participants that currently smoke, while our results indicate greater effectiveness for long-term former smokers. This is due to a difference in the granularity of former smoking behaviour considered between our study and the study of Pinsky and colleagues. While Pinsky and colleagues evaluated the effect of overall smoking status (current versus former), we evaluated the effect not only by smoking status, but also stratified by years since smoking cessation. Indeed, when we only consider stratification by smoking status without

considering years since smoking cessation, our results for NLST are consistent with those of Pinsky et al, as shown in the Figure (included as new Supplementary Figure S16) below:

We have further highlighted this in the discussion:

“Pinsky found heterogeneity in screening effectiveness by histology and potential heterogeneity by sex in NLST. However, their analysis considered “one-variable-at-a-time” rather than predictive approaches, precluding the identification of effect modification by patient characteristics. Our study confirms heterogeneity in screening effectiveness between histologies, but also accounted for confounding and effect modification by patient characteristics. For example, Pinsky et al found greater screening effectiveness for current smokers compared to former smokers. Our results are consistent when smoking status alone is considered (Supplementary Figure S16). However, our results indicate greater benefits for long-term former smokers when time since smoking cessation is also taken into consideration, demonstrating the importance of including sufficient granularity in former smoking behaviour.”

3) The overall impact of screening effectiveness by pack-years and former smoking. As demonstrated by the comparison to the results of Pinsky et al for former smoking behaviour, our results indicate sufficient granularity needs to be considered in interpreting the results. We believe the reviewer’s query with regards to screening effectiveness by pack-years primarily refers to the consistency between our findings and those of Kovalchik with regards to screening effectiveness by risk: as a greater number of pack-years is associated with a higher level of risk. As shown in parts 1) and 2) of our reply to this query, our results are consistent with those of both Kovalchik and Pinsky and colleagues. However, both pack-years and years since smoking cessation represent different components of overall risk: i.e. Person A

may have more pack-years than Person B, but also have quit longer than Person B. As a consequence, persons A and B may have a similar risk-level, but variation in their individual risk-factors. Our results provide insights in that while screening effectiveness may not vary by overall lung cancer risk (as demonstrated by Kovalchik et al), it does vary by individual risk-factors. This demonstrates the value of the comprehensive approach of our analyses. We have highlighted this important distinction in the discussion as follows:

“Our results indicate that while screening effectiveness does not vary by overall risk, it does vary across individual components of risk. Therefore, future studies should not only consider overall LC(M) risk, but also consider individual components of risk with sufficient granularity.”

Lastly, the analysis does not take into account the role of comorbid conditions and how they interact with lung cancer risk, histologic type, screening effectiveness and competing risks of death. There is an extensive emerging literature on this topic that is not cited or discussed.

We agree with the reviewer that further discussing the role of comorbid conditions is useful for the reader. Thus, we have included the following paragraph (along with relevant citations) in the discussion:

“Participants of both trials were more likely to be younger and have ceased smoking, but were generally representative of the general population meeting their inclusion criteria. Still, it is well known that the individuals eligible for lung cancer screening are more likely to have comorbidities such as COPD than those included in the trials. These comorbidities increase the overall risk of lung cancer, reduce life-expectancy and may affect both treatment effectiveness and the histological type of lung cancer that develops. Thus, future research should further evaluate the interplay between comorbid conditions and screening effectiveness.”

Minor comments:

Figure 1: this can be moved to the supplement. Instead, it would be helpful to include a Table with key characteristics of participants in the two trials.

We have moved Figure 1 to the supplement and replaced it with a Table with key characteristics of the participants in the two trials as suggested by the reviewer.

Figure 2: how were results pooled across methods?

The screening effectiveness estimates for each of the corresponding risk-factors and data points for Figure 2 were derived by taking the median of the point estimates for each individual method; i.e. the point estimate for the estimated screening effectiveness for the 50-54 year age group in NELSON was taken as the median of the point estimates of the rate-ratio, the effect model and the causal forest. Similarly, the medians of the upper and lower bounds of the confidence intervals of each method were used to construct the confidence intervals. We have clarified this as follows in the figure notes:

“The screening effectiveness estimates and 95% confidence intervals were based on the medians of the point estimates of the rate-ratios, effect models and causal forests. Similarly, the lower and upper bounds of the 95% confidence intervals were based on the medians of the lower and upper bounds across the considered methods. The estimates for the individual methods can be found in Supplementary Figures S6-S9.’

Figure 3: the outlier effect of adenocarcinoma histology when using the LLP model is potentially interesting and unsettling. What are some possible explanations?

We thank the reviewer for raising this point. As can be seen in Supplementary Tables S12 and S13, the second stage LLP model for adenocarcinoma-specific mortality in NLST is the only model that includes an interaction effect between the screening effectiveness parameter and the linear predictor of the first stage model. The figure demonstrates the estimate for the screening effectiveness parameter without the interaction effect. We have clarified this as follows in the figure notes:

“The adenocarcinoma-specific NLST estimate for the risk-model approach which uses LLPv3 risk in its first stage includes an interaction-effect between first-stage risk and screening effectiveness. The figure represents the estimate for the screening effectiveness parameter without the interaction effect.”

Reviewer #2 (Remarks to the Author):

The authors present a detailed comparison of the 2 largest positive lung cancer screening trials. Their findings are fairly presented and discussed and the conclusions are very largely supported by their findings. In addition, their discussion of the implications of their work and the trade-offs implied in the potential changes they suggest is valuable. A particular strength of the work is the comprehensive approach to the comparison, employing subgroup analyses, predictive modelling and machine learning approaches. The consistency of the findings is notable and adds confidence to the conclusions.

We thank the reviewer for their thoughtful suggestions. We have incorporated them as described in the next paragraphs.

The main determinant of screening effectiveness is the histological type of the lung cancer. Adenocarcinoma is more effectively detected than small cell lung cancer. This is discussed well between lines 286 and 294. Whilst various factors are considered, such as preclinical duration and smoking behaviour, location of the tumour is not discussed. This is an omission as adenocarcinoma tends to be peripheral and hence easily detected, whilst small cell lung cancer is central and can be more challenging to detect.

We agree that variations in tumour locality by histology is a factor that warrants further discussion. We have incorporated the suggestion of the reviewer as follows:

“Our findings are consistent with natural-history models that estimate longer preclinical durations and greater screen-detectability for histologies for which we find greater screening effectiveness. In particular, we find greater screening effectiveness for histologies that predominantly develop in locations that allow for easier detection through screening. For example, adenocarcinoma’s develop predominantly in the periphery of the lungs, as opposed to small-cell cancers that tend to be centrally located. Furthermore, our findings are consistent with regards to observed relations between histology and smoking behaviour, smoking behaviour and tumour growth within histologies and variations in survival by histology and smoking behaviour. Consequently, these mechanisms may drive the heterogeneity in screening effectiveness found in our study.”

The anomaly of squamous carcinoma detection effectiveness variance between the 2 studies is marked and attributed speculatively to semi-automated protocols being used in one study. This seems unlikely to account for such a large variance, which is worthy of further investigation. Is there any potential for confusion with non-malignant lesions, which might be better discriminated by automated protocols?

We thank the reviewer for highlighting this point. We believe this may be due to the difference in the detectability between overall tumour growth and volume doubling time between NELSON and NLST as described in our reply to reviewer #1. As such, we have included it as follows:

“While analyses based on NLST have suggested screening may not be beneficial for squamous-cell carcinoma, our analyses suggest it was beneficial in NELSON. This may be in part due to differences in nodule management protocols. Semi-automated measurements of nodule volume and volume doubling time as applied in NELSON has been shown to be more accurate in detecting nodule growth than the manual measurements of nodule diameter used in NLST. This is supported by a recent review that demonstrates that there may not be a significant change in volume at a three month follow-up scan, even when the volume doubling time is less than 400 days. Consequently, future studies should evaluate the impact of the differences in nodule management protocols between the trials on histology-specific screening effectiveness. ”

The authors make important suggestions for further work to investigate hypotheses generated in this work.

A few minor niggles:

Lines 58-60 could usefully be rephrased.

We have rephrased these lines for additional clarification as follows:

“Heterogeneity in LC screening effectiveness is primarily driven by histology. Relaxing smoking-related screening eligibility criteria may enhance screening effectiveness.”

Lines 171-174 discuss an important finding and it would be useful to have topline data in the main paper.

We agree with the reviewer and have added additional details as follows in the Results Section:

“Relative screening effectiveness was similar across baseline risk-quintiles for all risk-prediction models (Supplementary Figures S2-S5; NELSON medians: 22.7-24.6%, NLST medians: 8.9-13.4%) and second stage risk-models (Supplementary Table S5; NELSON median: 27.4%, NLST median: 15.6%). However, absolute screening benefits increased with quintile of baseline risk for all risk-prediction models (C-Outcomes Supplementary Appendix Figures C16-C19 NELSON LCM prevented per 1,000: Quintile 1 -0.92 to 2.31 to Quintile 5: 25.8-28.3, NLST LCM prevented per 1,000: Quintile 1 -0.04 to 1.03 to Quintile 5: 4.83-8.12).”

Lines 260-268 contains speculation on the results, more appropriate for the discussion.

We agree with the reviewer and have moved the paragraph on the potential causes for differences in LCM reductions between trials to the discussion.

Lines 407-414 make an important point, although the logical path from their findings should be clarified.

We agree with the reviewer that this paragraph would benefit from further clarification. We have rephrased this paragraph as follows:

“Integrating smoking cessation support has been shown to enhance the effectiveness of LC screening in reducing LCM through reducing the risk for developing LC. Our analyses suggest that integrating smoking cessation support further enhances the effectiveness of LC screening in reducing LCM through two additional pathways. Firstly, our analyses suggest that screening effectiveness is greater for former smokers compared to current smokers, particularly for long-term former smokers. Secondly, successful smoking cessation prevents the further accumulation of additional pack-years, which our analyses suggest is associated with reduced screening effectiveness. Consequently, the findings of our study may be used to further improve the uptake of smoking cessation services in LC screening programs.”

Reviewer #3 (Remarks to the Author):

This is an analysis of over 60,000 patients, pooled from two randomized screening trials (NELSON, NSLT), with the objective to assess the impact of patient heterogeneity on lung cancer screening effectiveness. Both standard approaches statistical methods and machine learning algorithms were considered to predict lung cancer mortality risk.

We thank the reviewer for their suggestions to further elucidate our findings to the reader.

Overall, the manuscript is hard to follow, as insufficient details are provided in the main text to understand the structure and goals of analyses and evaluate the findings. In general, I recommend to adhere more closely to TRIPOD statement guidelines and checklist (<https://www.tripod-statement.org/resources/>), but I provide specific comments below.

Although the reviewer suggests to adhere to the “Transparent reporting of a multivariable prediction model for individual prognosis or diagnosis (TRIPOD) statement” (Transparent Reporting of a multivariable prediction model for Individual Prognosis Or Diagnosis (TRIPOD): Explanation and Elaboration | Annals of Internal Medicine (acpjournals.org)), the main suggested guideline for the evaluation of heterogeneity in treatment effectiveness is the “Predictive Approaches to Treatment effect Heterogeneity (PATH) Statement” (The Predictive Approaches to Treatment effect Heterogeneity (PATH) Statement: Explanation and Elaboration | Annals of Internal Medicine), as mentioned in the introduction of our manuscript.

However, there is a degree of overlap in recommendations between the TRIPOD and PATH statements. As the TRIPOD statement has a checklist and the PATH statement (currently) does not, we have included a filled in version of the TRIPOD checklist to provide guidance for the reader. In addition to the

reviewer's specific comments, we have clarified the following specific items (in accordance with the checklist):

Title and abstract: We have added additional details in the abstract as suggested by the TRIPOD checklist.

Introduction: we have provided additional details on the rationale and objectives.

Methods: We have provided additional details on the data sources, participant characteristics, outcomes assessed, predictors used and creation of risk groups in the main text.

1. Please provide in the "Study population" section a detailed description of the arms of NELSON and NLST, including interventions and evaluated screening strategies.

We have added a more detailed description of the NELSON and NLST trials, including the interventions evaluated in each of their arms and the evaluated screening strategies as follows:

"In brief, the NELSON trial enrolled, from December 2003 through July 2006, individuals between the ages of 50-74, who smoked at least 15 cigarettes per day for ≥ 25 years or 10 cigarettes per day for ≥ 30 years and were current smokers or former smokers who quit < 10 years ago. The intervention arm received four rounds of computed tomography screening with different intervals: at baseline, year 1 (1-year interval), year 3 (2-year interval) and year 5.5 (2.5-year interval). The participants in the control arm received no screening. The NLST enrolled, from August 2002 through April 2004, individuals between the ages of 55-74, who smoked at least 30 pack-years and were current smokers or former smokers who quit < 15 years ago. The intervention arm received three rounds of computed tomography screening with a one-year interval each: at baseline, year 1 and year 2. The participants in the control arm received three rounds of chest radiography screening with the same schedule as the intervention arm."

In addition, as suggested by Reviewer #1, we have moved the original Figure 1 to the Supplement and replaced it with Table that provides a detailed overview of the characteristics of participants in both arms of the NELSON and NLST.

2. Better define in the methods the outcome that is predicted by all models/machine learning algorithms. "Lung cancer mortality" is too vague. How was this assessed? What is the time horizon for mortality predictions?

We have further defined the outcome and time horizon evaluated by the modelling methods as follows:

"Screening effectiveness was defined as the reduction in LCM achieved through CT screening. Screening effectiveness was evaluated by comparing the difference in LCM between each trial's screening and control group (NELSON: no screening; NLST: chest radiography screening) from randomisation until the end of each trial's respective follow-up period since randomization (10 years in NELSON and 7 years in NLST)."

3. Please explain or provide references explaining why the evaluation of LCM is not affected by lead-time/length-time bias.

Lead-time bias refers to the case where the time of detection of the disease is advanced due to screening, but the time of death remains unaltered. As survival is measured as the time between the

detection of the disease and the moment of death, screening artificially increases the survival duration for this case.

Length-time bias occurs due to screening favoring the detection of slower growing and less aggressive cancers, which have a better survival than more aggressive tumors. Thus, when one compares the survival rates of screen-detected cancers versus non-screen-detected cancers, screening may falsely appear to improve survival rates.

Cause-specific mortality is not affected by artificial increases in survival and is thus considered the gold standard for the primary outcome of screening trials. We have added a brief explanation, along with references, as follows:

“LCM was evaluated rather than survival, as survival estimates for screen-detected cases may be affected by lead-time and length-time bias.”

4. The "one-variable-at-the-time" analysis approach should be described in more detail in the main text. The Supplement describes provides a (unnecessary?) detailed technical description, but it is just based on exact univariable rate ratio tests based on the Poisson distribution.

We agree with the reviewer and have expanded the description as follows:

“Traditional sub-group analyses considered “one-variable-at-a-time” analyses, in which the trial population is serially divided into different groups (e.g. men and women) and the difference in LCM between groups is evaluated through exact univariable rate ratio tests based on the Poisson distribution.”

5. The definition of screening effectiveness should be clarified. In particular, it is not clear what is the specific effect that is being estimated (screening vs. no screening?) and what is the affect outcome variable (the risk predictions from the different models/algorithms? Why?)

The binary treatment intervention is CT screening versus the control arm (no screening in NELSON and CXR screening in NLST), while the affect outcome is a binary variable indicator stating whether the person died of lung cancer within the trial’s stated follow-up periods. We have further clarified the definition of screening effectiveness as follows:

“Screening effectiveness was defined as the reduction in LCM achieved through CT screening. LCM was evaluated rather than survival, as survival estimates for screen-detected cases may be affected by lead-time and length-time bias. Screening effectiveness was evaluated by comparing the difference in LCM between each trial’s screening and control group (NELSON: no screening; NLST: chest radiography screening) from randomisation until the end of each trial’s respective follow-up period since randomization (10 years in NELSON and 7 years in NLST).”

And the affect outcome variable as follows:

“All methods, except the rate-ratios, account for differences in participant characteristics between trials. Thus, the screening effectiveness parameter in these models represents the effect of screening on reducing LCM after accounting for confounders.”

REVIEWER COMMENTS

Reviewer #1 (Remarks to the Author):

The authors have addressed all of my questions. I very much appreciated their comprehensive and thoughtful response!

Reviewer #2 (Remarks to the Author):

The authors have addressed the reviewers' comments carefully and in detail. The changes they have made to the paper have significantly strengthened it.

Reviewer #3 (Remarks to the Author):

I thank the Authors for addressing my comments in a satisfactory manner. I have no further feedback to provide.

Author reply

We thank the reviewers for their thoughtful comments that helped us to revise the manuscript. We agree that they significantly strengthened the manuscript.